# EXACT LINEAR-RATE GRADIENT DESCENT: OPTIMAL ADAPTIVE STEPSIZE THEORY AND PRACTICAL USE

## ABSTRACT

Consider gradient descent iterations $\boldsymbol{x}^{k+1} = \boldsymbol{x}^k - \alpha_k \nabla f(\boldsymbol{x}^k)$. Suppose gradient exists and $\nabla f(\boldsymbol{x}^k) \neq \boldsymbol{0}$. We propose the following closed-form stepsize choice:

$$\alpha_k^\star = \frac{\|\boldsymbol{x}^\star - \boldsymbol{x}^k\|}{\|\nabla f(\boldsymbol{x}^k)\|} \cos \eta_k, \qquad \text{(theoretical)}$$

where $\eta_k$ is the angle between vectors $\boldsymbol{x}^\star - \boldsymbol{x}^k$ and $-\nabla f(\boldsymbol{x}^k)$. It is universally applicable and admits an exact linear convergence rate with factor $\sin^2 \eta_k$. Moreover, if $f$ is convex and $L$-smooth, then $\alpha_k^\star \geq 1/L$.

For practical use, we approximate (can be exact) the above via

$$\alpha_k^\dagger = \gamma_0 \cdot \frac{f(\boldsymbol{x}^k) - \bar{f}_0}{\|\nabla f(\boldsymbol{x}^k)\|^2}, \qquad \text{(practical use)}$$

where $\gamma_0$ is a tunable parameter; $\bar{f}_0$ is a guess on the smallest objective value (can be auto. updated). Suppose $f$ is convex and $\bar{f}_0 = f(\boldsymbol{x}^\star)$, then any choice from $\gamma_0 \in (0, 2]$ guarantees an exact linear-rate convergence to the optimal point.

We consider a few examples. (i) An $\mathbb{R}^2$ quadratic program, where a well-known ill-conditioning bottleneck is addressed, with a rate strictly better than $O(1/2^k)$. (ii) A geometric program, where an inaccurate guess $\bar{f}_0$ remains powerful. (iii) A non-convex MNIST classification problem via neural networks, where preliminary tests show that ours admits better performance than the state-of-the-art algorithms, particularly a tune-free version is available in some settings.

## 1 INTRODUCTION

The gradient descent (GD) algorithm, dated back to Cauchy in 1847, is arguably the most popular iterative algorithm. It is often treated as the default optimizer for neural networks Rumelhart et al. (1986); Ruder (2016); Goodfellow et al. (2016). GD's procedure is remarkably simple: repeatedly subtract the current iterate with its gradient. However, such a raw version suffers from a serious issue — it almost always overshoots the minimum. To guarantee convergence, damping the gradient by a stepsize $\alpha$ is necessary. How to properly choose such a stepsize is one of the most headache issues, since a large choice would overshoot and a small one leads to slow convergence. In practice, the stepsize (a.k.a. learning rate) is "often the single most important hyper-parameter" Bengio (2012).

To our best knowledge, in the current literature, a general convergence guarantee for GD only exists in the convex case, and requires at least one strong assumption, the $L$-smoothness. Specifically, if one can access the Lipschitz constant $L$, then any choice from $\alpha \in (0, 2/L)$ guarantees convergence, with $1/L$ the default choice, see e.g. (Ryu & Yin, 2022, Sec. 2.4.3). Despite such a guarantee being available, it is rarely used directly in large-scale problems, due to $L$ is either not computable or simply too expensive. There does exist some work that allow estimation of $L$, see e.g., Anil et al. (2019); Fazlyab et al. (2019); Combettes & Pesquet (2020). However, their focus is often not regarding the stepsize selection issue, appears related to the complication of the estimation scheme and that the estimation error in $L$ will propagate to the GD algorithm. In this manuscript, such an issue will be avoided, since our result does not rely on $L$.

One critique of the above classical theory is that the stepsize is fixed throughout all iterations of GD. This eliminates the possibility of some large feasible stepsize choices in the middle steps and

consequently slows down the algorithm. A better strategy should be adaptively adjusting the stepsize according to the current progress. Such an idea is old, at least traced back to Almeida et al. (1999). The real issue is how to adjust the stepsize adaptively? In the literature, several outstanding heuristic methods have been proposed, e.g., AdaGrad Duchi et al. (2011), RMSProp Tieleman & Hinton. (2012), Adam Kingma & Ba (2015). However, an adaptive stepsize theory has not been established. This manuscript will fill in this blank space. In the convex case, we show the feasible stepsize selection range that guarantees convergence being $(0, 2\alpha_k^\star)$, with $\alpha_k^\star$ the optimal $k$-th choice. Moreover, $\alpha_k^\star$ is lower bounded by $1/L$, implying the new range enlarges the aforementioned classical one $(0, 2/L)$. Also, our optimal stepsize yields an exact linear rate with factor $\sin^2 \eta_k$. Let us note that if $\sin \eta_k = 0$, then GD will converge instantly, see an example in Section 4.1.2.

Remarkably, our theory also applies to a non-convex function. A notable difference is that the optimal choice $\alpha_k^\star$ can be negative now, and the feasible range becomes either $(2\alpha_k^\star, 0)$ or $(0, 2\alpha_k^\star)$, depending on the sign of $\alpha_k^\star$. The negative sign is not too surprising, since if the function is locally concave, we do need an ascent direction to pass the hill, otherwise stuck at the local minimum. This aspect shares a similar flavour to the so-called 'gradient descent ascent' method for solving min-max problems, see e.g. Lin et al. (2020); Zheng et al. (2024).

Despite our non-convex applicability, the situation is highly challenging. Unlike the convex case where $\alpha_k^\star$ is lower bounded, here it can take an arbitrary value. The worst case is when $\alpha_k^\star = 0$, implying an empty selection range. This arises when $\boldsymbol{x}^\star - \boldsymbol{x}^k \perp \nabla f(\boldsymbol{x}^k)$, and a stepsize that can improve the current iterate $\boldsymbol{x}^k$ does not exist. On the other hand, if one can exclude such an orthogonal case, then convergence to the global optimal point is guaranteed, see Theorem 2.1.

While our theory is powerful, it is not instantly useful in practice, due to quantity $\langle \boldsymbol{x}^\star - \boldsymbol{x}^k, -\nabla f(\boldsymbol{x}^k) \rangle$ is not a priori knowledge. Experts may instantly realize that, by Taylor expansion, it is an upper bound for $f(\boldsymbol{x}^k) - f(\boldsymbol{x}^\star)$ in the convex case, and the only concern is regarding $f(\boldsymbol{x}^\star)$. We show that,

(i) when $f(\boldsymbol{x}^\star) = 0$, the simplest tune-free stepsize $f(\boldsymbol{x}^k)/\|\nabla f(\boldsymbol{x}^k)\|^2$ is applicable. It is at least $1/(2L)$ large, see Proposition 3.2. In a special case, its two-times scaled version is optimal, see Section 3.1.3.

(ii) when $f(\boldsymbol{x}^\star)$ not known in advance, a parameter $\bar{f}_0$ is introduced as an initial guess for $f(\boldsymbol{x}^\star)$. It will be updated if some criteria violated, see details in Algorithm 1. Moreover, such a guess can be easily picked, for example, let $\bar{f}_0 = 0.1 \cdot f(\boldsymbol{x}^0)$, where $f(\boldsymbol{x}^0)$ is the initial objective value.

An outstanding benefit of our scheme is regarding the ill-conditioning issue, which is a well-known bottleneck for the GD algorithm. This aspect has been nicely illustrated in (Boyd & Vandenberghe, 2004, Sec. 9.3.2) through an $\mathbb{R}^2$ example, where an exact linear rate with factor $(\gamma - 1)^2/(\gamma + 1)^2$ is given, using an exact line search stepsize. A large $\gamma$ (ill-conditioning) causes such a factor close to 1, implying the error has almost no change as GD iterating. Ours yields a factor of $(\gamma - 1)^2/(2\gamma^2 + 2)$, which is strictly smaller than $1/2$, i.e., the error is at least halved each iteration, see more details in Section 4.1.

For notations, $\| \cdot \|$ denotes the Euclidean norm, induced by the inner product $\langle \cdot, \cdot \rangle$. The uppercase bold, lowercase bold, and not bold letters are used for matrices, vectors, and scalars, respectively.

## 1.1 LITERATURE: ADAPTIVE STEPSIZE

Here, we briefly discuss some developments of the stepsize adaption technique in the machine learning field. The most popular family includes AdaGrad Duchi et al. (2011), RMSProp Tieleman & Hinton. (2012), and Adam Kingma & Ba (2015). These approaches are strongly related to each other and are heuristic methods that typically require tuning multiple parameters. Recently, Baydin et al. (2018) propose to adaptively update the stepsize via a so-called 'hyper-gradient', which computes a derivative over the stepsize parameter. The good news is that doing so adds very limited cost owing to an element-wise product. The bad news is that the 'hyper-gradient' introduces a 'hyper-stepsize' which still needs tuning (but tends to be easier). Also, a theoretical convergence guarantee is not yet available. A follow-up work by Chandra et al. (2022) addresses the tuning issue by computing an additional 'hyper-gradient' on the original 'hyper-stepsize'. This would introduce another 'hyper-stepsize', and they apply the same procedure again, and so on, ad infinitum. The good news is that each additionally introduced 'hyper-gradient' reduces the stepsize sensitivity, and eventually they can easily pick an initial hyper-stepsize.

In view of these methods, we note that there is always an initial stepsize tuning issue, also referred to as 'the global learning rate' selection. This issue is avoided in our approach, since all of our choices, including the initial one, are mathematically computed.

## 1.2 KEY RESULTS

Below, we summarize 3 versions of our adaptive stepsize choices.

- (i) Theoretically, the $k$-th optimal choice

$$\alpha_k^{\star} = \frac{\langle \boldsymbol{x}^{\star} - \boldsymbol{x}^k, -\nabla f(\boldsymbol{x}^k) \rangle}{\|\nabla f(\boldsymbol{x}^k)\|^2} = \frac{\|\boldsymbol{x}^{\star} - \boldsymbol{x}^k\|}{\|\nabla f(\boldsymbol{x}^k)\|} \cos \eta_k, \quad k = 0, 1, \dots, \tag{1.1}$$

where $\eta_k = \arccos \frac{\langle \boldsymbol{x}^{\star} - \boldsymbol{x}^k, -\nabla f(\boldsymbol{x}^k) \rangle}{\|\boldsymbol{x}^{\star} - \boldsymbol{x}^k\| \|\nabla f(\boldsymbol{x}^k)\|}$. It admits an exact linear rate, with or without convexity:

$$\|\boldsymbol{x}^{k+1} - \boldsymbol{x}^{\star}\|^2 = \left( \Pi_{t=0}^k \sin^2 \eta_t \right) \|\boldsymbol{x}^0 - \boldsymbol{x}^{\star}\|^2. \tag{1.2}$$

- (ii) The $k$-th practical-use choice (general version)

$$\alpha_k^{\dagger} = \gamma_0 \cdot \frac{f(\boldsymbol{x}^k) - \bar{f}_0}{\|\nabla f(\boldsymbol{x}^k)\|^2}, \tag{1.3}$$

which admits the following exact linear rate, with or without convexity:

$$\|\boldsymbol{x}^{k+1} - \boldsymbol{x}^{\star}\|^2 = \left( \Pi_{t=0}^k \delta_t \right) \|\boldsymbol{x}^0 - \boldsymbol{x}^{\star}\|^2, \qquad k = 0, 1, \dots, \tag{1.4}$$

where

$$\delta_t = 1 - \frac{\gamma_0}{\sigma_t} \left( 2 - \frac{\gamma_0}{\sigma_t} \right) \cos^2 \eta_t, \qquad \sigma_t = \frac{\langle \boldsymbol{x}^{\star} - \boldsymbol{x}^t, -\nabla f(\boldsymbol{x}^t) \rangle}{f(\boldsymbol{x}^t) - \bar{f}_0}. \tag{1.5}$$

- (iii) The simplest practical-use choice (tune-free)

$$\widetilde{\alpha}_k = \frac{f(\boldsymbol{x}^k)}{\|\nabla f(\boldsymbol{x}^k)\|^2}, \tag{1.6}$$

which guarantees convergence if $f$ is convex and $f(\boldsymbol{x}^{\star}) = 0$. Empirically, it also works nicely for the non-convex MNIST problem in some settings.

# 2 ADAPTIVE STEPSIZE THEORY

Consider the following problem:

$$\underset{\boldsymbol{x} \in \mathbb{R}^n}{\text{minimize}} \ f(\boldsymbol{x}), \tag{2.1}$$

where function $f : \mathbb{R}^n \to \mathbb{R}$ is assumed to be everywhere differentiable. The associated gradient descent (GD) iterates are

$$\boldsymbol{x}^{k+1} = \boldsymbol{x}^k - \alpha_k \nabla f(\boldsymbol{x}^k), \quad k = 0, 1, \dots. \tag{2.2}$$

Throughout the rest of the paper, we assume $\nabla f(\boldsymbol{x}^k) \neq \boldsymbol{0}$, unless GD already converged $\boldsymbol{x}^k = \boldsymbol{x}^{\star}$. This assumption is necessary, since otherwise GD yields $\boldsymbol{x}^{k+1} = \boldsymbol{x}^k - \alpha_k \cdot \boldsymbol{0} = \boldsymbol{x}^k$, and the stepsize selection issue becomes trivial.

## 2.1 SELECTION RANGE

First, we show a feasible selection range for stepsize $\alpha$ to guarantee convergence.

**Proposition 2.1** (range). *Consider GD in equation 2.2. While iterates not converged, let stepsize*

$$\alpha_k \in \left( \frac{2\langle \boldsymbol{x}^{\star} - \boldsymbol{x}^k, -\nabla f(\boldsymbol{x}^k) \rangle}{\|\nabla f(\boldsymbol{x}^k)\|^2}, 0 \right) \bigcup \left( 0, \frac{2\langle \boldsymbol{x}^{\star} - \boldsymbol{x}^k, -\nabla f(\boldsymbol{x}^k) \rangle}{\|\nabla f(\boldsymbol{x}^k)\|^2} \right), \quad k = 0, 1, \dots \tag{2.3}$$

*If such $\alpha_k$ exists $\forall k$. Then, convergence to the global optimal point is guaranteed.*

**Corollary 2.1.** *$\alpha_k$ as in equation 2.3 does not exist if and only if*

$$\langle \boldsymbol{x}^{\star} - \boldsymbol{x}^k, -\nabla f(\boldsymbol{x}^k) \rangle = 0. \tag{2.4}$$

*Remarks* 2.1 (interpretation). In view of Corollary 2.1, it says that a feasible stepsize does not exist, if vectors $\boldsymbol{x}^{\star} - \boldsymbol{x}^k$ and $-\nabla f(\boldsymbol{x}^k)$ are orthogonal (zero vector case omitted by assumption). This is not surprising, since when orthogonality arises, by changing stepsize $\alpha_k$ alone, the future iterate $\boldsymbol{x}^{k+1} = \boldsymbol{x}^k - \alpha_k \nabla f(\boldsymbol{x}^k)$ cannot be any closer to $\boldsymbol{x}^{\star}$ than that of $\boldsymbol{x}^k$.

## 2.2 OPTIMAL CHOICE

Here, we present the optimal stepsize choice from the above feasible range. It turns out to be its central point.

**Theorem 2.1** (optimal choice)**.** *Consider GD in equation 2.2. The optimal $k$-th choice is given by*

$$\alpha_k^\star = \frac{\langle \boldsymbol{x}^\star - \boldsymbol{x}^k, -\nabla f(\boldsymbol{x}^k)\rangle}{\|\nabla f(\boldsymbol{x}^k)\|^2} = \frac{\|\boldsymbol{x}^\star - \boldsymbol{x}^k\|}{\|\nabla f(\boldsymbol{x}^k)\|}\cos\eta_k, \tag{2.5}$$

*where $\eta_k \stackrel{\text{def}}{=} \arccos \frac{\langle \boldsymbol{x}^\star - \boldsymbol{x}^k, -\nabla f(\boldsymbol{x}^k)\rangle}{\|\boldsymbol{x}^\star - \boldsymbol{x}^k\|\|\nabla f(\boldsymbol{x}^k)\|}$. It admits the following exact adaptive linear rate:*

$$\|\boldsymbol{x}^{k+1} - \boldsymbol{x}^\star\|^2 = \left(\Pi_{t=0}^k \sin^2 \eta_t\right)\|\boldsymbol{x}^0 - \boldsymbol{x}^\star\|^2, \quad k = 0, 1, \ldots. \tag{2.6}$$

*Remarks* 2.2 (scaling invariance)**.** GD equipped with $\alpha_k^\star$ in equation 2.5 is invariant under a linearly transformed function, $g(\cdot) = \rho f(\cdot), \forall \rho \neq 0$, since

$$\boldsymbol{x}^{k+1} = \boldsymbol{x}^k - \frac{\langle \boldsymbol{x}^\star - \boldsymbol{x}^k, -\rho\nabla f(\boldsymbol{x}^k)\rangle}{\|\rho\nabla f(\boldsymbol{x}^k)\|^2}\rho\nabla f(\boldsymbol{x}^k) = \boldsymbol{x}^k - \alpha_k^\star \nabla f(\boldsymbol{x}^k). \tag{2.7}$$

## 2.3 CONVEXITY

Suppose function $f$ is convex. Then, much stronger guarantees and simplifications are available.

**Corollary 2.2.** *Consider GD in equation 2.2. Suppose function $f$ is convex. While iterates not converged, let stepsize*

$$\alpha_k \in \left(0, \frac{2\langle \boldsymbol{x}^\star - \boldsymbol{x}^k, -\nabla f(\boldsymbol{x}^k)\rangle}{\|\nabla f(\boldsymbol{x}^k)\|^2}\right) = (0, 2\alpha_k^\star), \quad k = 0, 1, \ldots. \tag{2.8}$$

*Then, the GD iterations are guaranteed to converge to the optimal point.*

*Remarks* 2.3**.** Given a convex function $f$, relation $\langle \boldsymbol{x}^\star - \boldsymbol{x}^k, -\nabla f(\boldsymbol{x}^k)\rangle > 0$ always holds, unless $\boldsymbol{x}^k = \boldsymbol{x}^\star$.

### 2.3.1 L-SMOOTH

Here, we provide some characterizations via the $L$-smoothness assumption.

**Definition 2.1.** *A differentiable convex function $f : \mathbb{R}^n \to \mathbb{R}$ is said to be L-smooth if*

$$\|\nabla f(\boldsymbol{x}) - \nabla f(\boldsymbol{y})\| \leq L\|\boldsymbol{x} - \boldsymbol{y}\|, \qquad \forall \boldsymbol{x}, \boldsymbol{y} \in \mathbb{R}^n. \tag{2.9}$$

**Proposition 2.2.** *Suppose function $f : \mathbb{R}^n \to \mathbb{R}$ is L-smooth. Then,*

$$\alpha_k^\star = \frac{\langle \boldsymbol{x}^\star - \boldsymbol{x}^k, -\nabla f(\boldsymbol{x}^k)\rangle}{\|\nabla f(\boldsymbol{x}^k)\|^2} \geq \frac{1}{L}, \quad k = 0, 1.... \tag{2.10}$$

**Corollary 2.3.** *The fixed stepsize selection range is a subset of our adaptive one, i.e.,*

$$\left(0, \frac{2}{L}\right) \subseteq (0, 2\alpha_k^\star), \quad k = 0, 1.... \tag{2.11}$$

# 3 PRACTICAL USE

The above theory involves optimal point $\boldsymbol{x}^\star$, hence not instantly useful in practice. Here, we address it via approximation.

**Theorem 3.1.** *Consider GD in equation 2.2. While iterates not converged, we propose stepsize*

$$\alpha_k^\dagger = \gamma_0 \cdot \frac{f(\boldsymbol{x}^k) - \bar{f}_0}{\|\nabla f(\boldsymbol{x}^k)\|^2}, \tag{3.1}$$

*where $\gamma_0$ is a tunable parameter; $\bar{f}_0$ is a guessed smallest objective value. It admits the following exact linear rate:*

$$\|\boldsymbol{x}^{k+1} - \boldsymbol{x}^\star\|^2 = \left(\Pi_{t=0}^k \delta_t\right)\|\boldsymbol{x}^0 - \boldsymbol{x}^\star\|^2, \tag{3.2}$$

*where*

$$\delta_t = 1 - \frac{\gamma_0}{\sigma_t}\left(2 - \frac{\gamma_0}{\sigma_t}\right)\cos^2 \eta_t, \qquad \sigma_t = \frac{\langle \boldsymbol{x}^\star - \boldsymbol{x}^t, -\nabla f(\boldsymbol{x}^t)\rangle}{f(\boldsymbol{x}^t) - \bar{f}_0}, \tag{3.3}$$

*and where $\eta_t = \arccos \frac{\langle \boldsymbol{x}^\star - \boldsymbol{x}^t, -\nabla f(\boldsymbol{x}^t)\rangle}{\|\boldsymbol{x}^\star - \boldsymbol{x}^t\|\|\nabla f(\boldsymbol{x}^t)\|}$.*

**Corollary 3.1** (convergence). *While GD iterates not converged, let*

$$\gamma_0 \in (2\sigma_k, 0) \cup (0, 2\sigma_k), \quad k = 0, 1, \dots. \tag{3.4}$$

*If such $\gamma_0$ exists $\forall k$, then*

$$\delta_k = 1 - \frac{\gamma_0}{\sigma_k} \left(2 - \frac{\gamma_0}{\sigma_k}\right) \cos^2 \eta_k \in (0, 1), \quad \forall k, \tag{3.5}$$

*which guarantees convergence to the global optimal point.*

**Corollary 3.2.** *The optimal $k$-th choice of the tunable parameter $\gamma_0$ is*

$$\gamma_0^\star = \underset{\gamma_0}{argmax} \; \frac{\gamma_0}{\sigma_k} \left(2 - \frac{\gamma_0}{\sigma_k}\right) = \sigma_k. \tag{3.6}$$

*In this case, the rate factor*

$$\delta_k^\star = 1 - \frac{\gamma_0^\star}{\sigma_k} \left(2 - \frac{\gamma_0^\star}{\sigma_k}\right) \cos^2 \eta_k = \sin^2 \eta_k, \tag{3.7}$$

*implying optimality attained (recall Theorem 2.1), i.e., exact approximation.*

*Remarks* 3.1. In view of Corollary 3.2, the approximation is exact if one can adaptively select $\gamma_0 = \sigma_k, \forall k$. There does exist a special case where $\sigma_k$ is a known constant, see Section 3.1.3. However, in general, we do not know $\sigma_k$ in advance, and our approximation hence not exact. Also, for ease of use, we typically fix $\gamma_0$ to be a constant, which is theoretically sub-optimal.

## 3.1 CONVEXITY

Suppose function $f$ is convex. Then, we have stronger guarantees and a tune-free stepsize selection scheme.

**Corollary 3.3** (convergence). *Suppose function $f$ is convex. While GD iterates not converged, let*

$$\gamma_0 \in (0, 2\sigma_k), \quad \forall k, \tag{3.8}$$

*where $\sigma_k = \frac{\langle \boldsymbol{x}^\star - \boldsymbol{x}^k, -\nabla f(\boldsymbol{x}^k) \rangle}{f(\boldsymbol{x}^k) - \bar{f}_0}$. Then, the rate factor satisfies*

$$\delta_k = 1 - \frac{\gamma_0}{\sigma_k} \left(2 - \frac{\gamma_0}{\sigma_k}\right) \cos^2 \eta_k \in (0, 1), \quad \forall k, \tag{3.9}$$

*which guarantees convergence.*

### 3.1.1 TUNE-FREE CASE

Here, we require full knowledge of $f(\boldsymbol{x}^\star)$.

**Proposition 3.1.** *Consider GD in equation 2.2. Suppose function $f$ is convex, with optimal objective value $f(\boldsymbol{x}^\star)$ known in advance. Then, stepsize*

$$\widetilde{\alpha}_k = \gamma_0 \cdot \frac{f(\boldsymbol{x}^k) - f(\boldsymbol{x}^\star)}{\|\nabla f(\boldsymbol{x}^k)\|^2}, \quad \gamma_0 \in (0, 2], \tag{3.10}$$

*guarantees convergence, with an exact linear rate:*

$$\|\boldsymbol{x}^{k+1} - \boldsymbol{x}^\star\|^2 = \left(\Pi_{t=0}^k \; \delta_t\right) \|\boldsymbol{x}^0 - \boldsymbol{x}^\star\|^2, \tag{3.11}$$

*where*

$$\delta_t = 1 - \frac{\gamma_0}{\sigma_t} \left(2 - \frac{\gamma_0}{\sigma_t}\right) \cos^2 \eta_t, \quad \sigma_t = \frac{\langle \boldsymbol{x}^\star - \boldsymbol{x}^t, -\nabla f(\boldsymbol{x}^t) \rangle}{f(\boldsymbol{x}^t) - f(\boldsymbol{x}^\star)}, \tag{3.12}$$

*where $\eta_t = \arccos \frac{\langle \boldsymbol{x}^\star - \boldsymbol{x}^t, -\nabla f(\boldsymbol{x}^t) \rangle}{\|\boldsymbol{x}^\star - \boldsymbol{x}^t\| \|\nabla f(\boldsymbol{x}^t)\|}$.*

*Remarks* 3.2. The above tune-free case can happen in practice. A typical example is when $f(\boldsymbol{x}^\star) = 0$, arising in (i) solving a huge-scale linear system $\boldsymbol{A}\boldsymbol{x} = \boldsymbol{b}$, where $\boldsymbol{A}^{-1}$ is too expensive to calculate directly; (ii) $f$ is a loss function with zero-loss at the optimal point, as in many classification problems.

**Corollary 3.4.** *Suppose $f$ is a non-linear convex function. Then, when $\boldsymbol{x}^k \neq \boldsymbol{x}^\star$, we have*

$$\gamma_0^\star = \sigma_k = \frac{\langle \boldsymbol{x}^\star - \boldsymbol{x}^k, -\nabla f(\boldsymbol{x}^k) \rangle}{f(\boldsymbol{x}^k) - f(\boldsymbol{x}^\star)} > 1, \quad \forall k. \tag{3.13}$$

*Remarks* 3.3. equation 3.13 follows instantly from Taylor expansion. It implies that we should choose $\gamma_0 > 1$ in our convex tune-free case. However, it does not tell exactly how much larger than 1, our default choice is therefore conservatively set to $\gamma_0 = 1$. Additionally, we assume $f$ being non-linear, since minimizing a linear or affine function is trivial (unbounded below).

### 3.1.2 $L$-SMOOTH

Here, we provide some characterizations via the $L$-smooth assumption.

**Proposition 3.2.** *Suppose function $f : \mathbb{R}^n \to \mathbb{R}$ is L-smooth. Then,*

$$\frac{f(\boldsymbol{x}^k) - f(\boldsymbol{x}^\star)}{\|\nabla f(\boldsymbol{x}^k)\|^2} \geq \frac{1}{2L}, \tag{3.14}$$

**Proposition 3.3** (optimality gap). *Let function $f : \mathbb{R}^n \to \mathbb{R}$ be L-smooth. Then,*

$$\underbrace{\frac{\langle \boldsymbol{x}^\star - \boldsymbol{x}^k, -\nabla f(\boldsymbol{x}^k) \rangle}{\|\nabla f(\boldsymbol{x}^k)\|^2}}_{optimal} - \underbrace{\frac{f(\boldsymbol{x}^k) - f(\boldsymbol{x}^\star)}{\|\nabla f(\boldsymbol{x}^k)\|^2}}_{estimated\,(\gamma_0 = 1)} \geq \frac{1}{2L}. \tag{3.15}$$

*Remarks* 3.4. The positive gap from Proposition 3.3 with $\gamma_0 = 1$ is not surprising, since we already seen from Corollary 3.4 that the optimal parameter $\gamma_0^\star$ is strictly larger than 1 (and $\gamma_0^\star$ attains optimality by Corollary 3.2). The result here is strengthened, with the gap characterized by $L$, instead of only being positive.

### 3.1.3 PRACTICAL EXACT APPROXIMATION

Here, we show special cases that our practical-use stepsize choice attains the theoretical optimum, by simply selecting $\gamma_0 = 2$. Consider

$$\underset{\boldsymbol{x} \in \mathbb{R}^n}{\text{minimize}} \quad \frac{1}{2}\|\boldsymbol{A}\boldsymbol{x} - \boldsymbol{b}\|^2. \tag{3.16}$$

where $\boldsymbol{x} \in \mathbb{R}^n, \boldsymbol{b} \in \mathbb{R}^m, \boldsymbol{A} \in \mathbb{R}^{m \times n}$.

(i) Suppose $\boldsymbol{A}$ is a full-rank square matrix. We have $\boldsymbol{x}^\star = \boldsymbol{A}^{-1}\boldsymbol{b}$. It follows that,

$$\alpha_k^\star = \frac{\langle \boldsymbol{x}^\star - \boldsymbol{x}^k, -\nabla f(\boldsymbol{x}^k) \rangle}{\|\nabla f(\boldsymbol{x}^k)\|^2} = \frac{\langle \boldsymbol{A}^{-1}\boldsymbol{b} - \boldsymbol{x}^k, -\boldsymbol{A}^T(\boldsymbol{A}\boldsymbol{x}^k - \boldsymbol{b}) \rangle}{\|\nabla f(\boldsymbol{x}^k)\|^2} = \frac{\|\boldsymbol{A}\boldsymbol{x}^k - \boldsymbol{b}\|^2}{\|\nabla f(\boldsymbol{x}^k)\|^2} = \frac{2 \cdot f(\boldsymbol{x}^k)}{\|\nabla f(\boldsymbol{x}^k)\|^2}, \tag{3.17}$$

corresponding to our practical-use stepsize with $\gamma_0 = 2$ and $f(\boldsymbol{x}^\star) = 0$, recall equation 3.10.

(ii) Suppose $\boldsymbol{b} = \boldsymbol{0}$. We have $\boldsymbol{x}^\star = \boldsymbol{0}$. It follows that,

$$\alpha_k^\star = \frac{\langle \boldsymbol{x}^\star - \boldsymbol{x}^k, -\nabla f(\boldsymbol{x}^k) \rangle}{\|\nabla f(\boldsymbol{x}^k)\|^2} = \frac{\langle \boldsymbol{0} - \boldsymbol{x}^k, -\boldsymbol{A}^T(\boldsymbol{A}\boldsymbol{x}^k - \boldsymbol{0}) \rangle}{\|\nabla f(\boldsymbol{x}^k)\|^2} = \frac{\|\boldsymbol{A}\boldsymbol{x}^k\|^2}{\|\nabla f(\boldsymbol{x}^k)\|^2} = \frac{2 \cdot f(\boldsymbol{x}^k)}{\|\nabla f(\boldsymbol{x}^k)\|^2}, \tag{3.18}$$

which is similar to the above case.

### 3.2 GENERAL PRACTICAL USE ALGORITHM

Here, we consider $\bar{f}_0$ being an inaccurate guess. It will be updated if certain criteria violated.

---

**Algorithm 1** Linear-rate gradient decent (auto correction version)

---

**Input:** initialization $\boldsymbol{x}^0$; iteration number counter $k = 0$;
**Input:** guessed $\bar{f}_0$, tunable parameter $\gamma_0$;
**Input:** shrinking factors $\tau_1, \tau_2 \in (0, 1)$, threshold $T$.
 1: **while** iterates not converged **do**
 2: $\quad k \;\leftarrow\; k + 1$
 3: $\quad \alpha_k \;\leftarrow\; \gamma_0 \cdot \frac{f(\boldsymbol{x}^k) - \bar{f}_0}{\|\nabla f(\boldsymbol{x}^k)\|^2}$,
 4: $\quad \boldsymbol{x}^{k+1} \;\leftarrow\; \boldsymbol{x}^k - \alpha_k \nabla f(\boldsymbol{x}^k)$
 5: $\quad$ **Correction**:
 $\qquad$ If $f(\boldsymbol{x}^{k+1}) > T \cdot f(\boldsymbol{x}^k)$, set $\gamma_0 \leftarrow \tau_1 \cdot \gamma_0$ and $\boldsymbol{x}^{k+1} \leftarrow \boldsymbol{x}^k$.
 $\qquad$ If $\alpha_k \leq 0$, set $\bar{f}_0 \leftarrow \tau_2 \cdot \bar{f}_0$.
 6: **end while**
**Output:** $\boldsymbol{x}^{k+1}$

---

## 4  EXAMPLES

In this section, through some specific examples, we demonstrate the power of our adaptive stepsize.

### 4.1  $\mathbb{R}^2$ QUADRATIC PROGRAM

Here, we consider a simple example from (Boyd & Vandenberghe, 2004, Sec. 9.3.2):

$$\underset{x_1, x_2}{\text{minimize}} \ \frac{1}{2} \left( x_1^2 + \gamma x_2^2 \right), \tag{4.1}$$

where $\gamma > 0$, $\boldsymbol{x} = [x_1, x_2]^T$. We employ initialization $\boldsymbol{x}^0 = [\gamma, 1]^T$. For this problem, the Lipschitz constant $L$ and the condition number both equal to $\gamma$, and the conditioning state is fully tractable.

Below, we compare our approach with the exact line search method, which finds a stepsize choice via

$$\alpha_k = \underset{\alpha_k > 0}{\operatorname{argmin}} \ f(\boldsymbol{x}^k - \alpha_k \nabla f(\boldsymbol{x}^k)). \tag{4.2}$$

• Following from (Boyd & Vandenberghe, 2004, Sec. 9.3.2), the $k$-th iterate with an exact line search stepsize is given by

$$x_1^k = \gamma \left( \frac{\gamma - 1}{\gamma + 1} \right)^k, \quad x_2^k = \left( -\frac{\gamma - 1}{\gamma + 1} \right)^k, \tag{4.3}$$

with an exact convergence rate

$$\frac{\|\boldsymbol{x}^k - \boldsymbol{x}^\star\|^2}{\|\boldsymbol{x}^0 - \boldsymbol{x}^\star\|^2} = \left( \frac{\gamma - 1}{\gamma + 1} \right)^{2k}. \tag{4.4}$$

If $\gamma$ is large (ill-conditioning), the above factor is close to 1, i.e., $\|\boldsymbol{x}^k - \boldsymbol{x}^\star\|^2$ is similar to $\|\boldsymbol{x}^0 - \boldsymbol{x}^\star\|^2$. That said, the ground-truth error has little changes after $k$ iterations.

• Our optimal choice $\alpha_k^\star$ yields

$$x_1^k = \gamma^{k_2 - k_1 + 1} \left( \frac{\gamma - 1}{2} \right)^{k_1} \left( \frac{\gamma - 1}{\gamma^2 + 1} \right)^{k_2},$$

$$x_2^k = (-1)^{k_1 + k_2} \left( \frac{\gamma - 1}{2} \right)^{k_1} \left( \frac{\gamma - 1}{\gamma^2 + 1} \right)^{k_2}, \tag{4.5}$$

where $k_1 \overset{\text{def}}{=} \lfloor \frac{k+1}{2} \rfloor$, $k_2 \overset{\text{def}}{=} \lfloor \frac{k}{2} \rfloor$, and where $\lfloor \cdot \rfloor$ denotes the floor operation (the closest smaller integer). Ours admits the following convergence rate factor:

$$\frac{\|\boldsymbol{x}^k - \boldsymbol{x}^\star\|^2}{\|\boldsymbol{x}^0 - \boldsymbol{x}^\star\|^2} = \frac{\gamma^{2(k_2 - k_1 + 1)} + 1}{\gamma^2 + 1} \left( \frac{\gamma - 1}{2} \right)^{2k_1} \left( \frac{\gamma - 1}{\gamma^2 + 1} \right)^{2k_2} = \left( \frac{1}{2} \right)^k \left( \frac{\gamma - 1}{\sqrt{\gamma^2 + 1}} \right)^{2k}. \tag{4.6}$$

Since $\gamma > 0$, we have $(\gamma - 1)/\sqrt{\gamma^2 + 1} < 1$. Our factor is therefore strictly smaller than $1/2^k$.

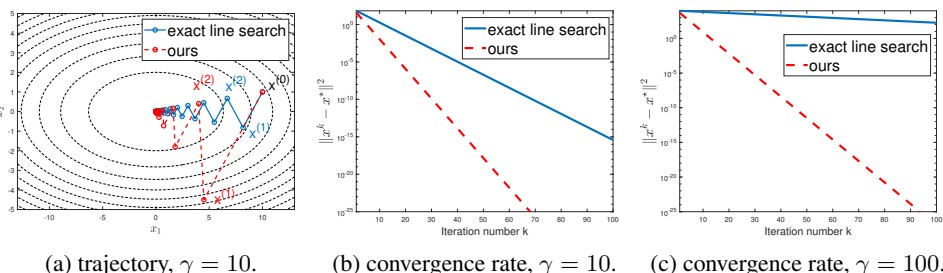

(a) trajectory, $\gamma = 10$.  (b) convergence rate, $\gamma = 10$.  (c) convergence rate, $\gamma = 100$.

Figure 1: exact line search vs. our stepsize, with conditioning controlled by $\gamma$.

#### 4.1.1 STRICT BETTER PERFORMANCE

Here, we show that our rate is strictly better than the one for exact line search, except when $\gamma = 1$ both methods converge in exactly one iteration.

To this end, suppose $\gamma \neq 1$. Divide our rate factor by that of the exact line search, we arrive at

$$\frac{\delta_{\text{ours}}^{(k)}}{\delta_{\text{line-search}}^{(k)}} = \left(\frac{1}{2}\right)^k \left(\frac{\gamma-1}{\sqrt{\gamma^2+1}}\right)^{2k} \left(\frac{\gamma+1}{\gamma-1}\right)^{2k} = \left(\frac{\gamma^2+2\gamma+1}{2\gamma^2+2}\right)^k < 1, \quad (4.7)$$

where the last inequality follows from the denominator being larger when $\gamma \neq 1$, since

$$2\gamma^2 + 2 - (\gamma^2 + 2\gamma + 1) = \gamma^2 - 2\gamma + 1 = (\gamma - 1)^2 > 0. \quad (4.8)$$

#### 4.1.2 INSTANT CONVERGENCE

Here, we perform an additional test on our rate factor $\sin^2 \eta_k$. An observation is that if $\sin \eta_k = 0$, GD must converge instantly. In the current example, we can easily verify it using a sparse initialization, say $(x_1^0, x_2^0) = (50, 0)$. Indeed, 1-step instant convergence is observed.

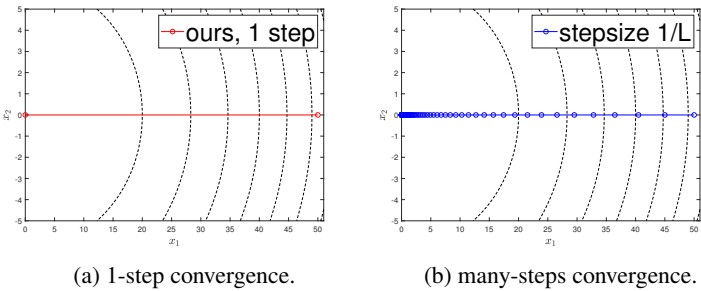

(a) 1-step convergence.  (b) many-steps convergence.

Figure 2: zero angle case, $\gamma = 10$.

### 4.2 GEOMETRIC PROGRAM

Here, we consider an unconstrained geometric program from (Boyd & Vandenberghe, 2004, Sec. 9.3), and our Algorithm 1 will apply. Consider

$$\underset{\boldsymbol{x}}{\text{minimize}} \ \log \left(\sum_{i=1}^{m} \exp\left(\boldsymbol{a}_i^T \boldsymbol{T} \boldsymbol{x} + b_i\right)\right), \quad (4.9)$$

where $\boldsymbol{x} \in \mathbb{R}^n$, $\boldsymbol{a}_i \in \mathbb{R}^n$, $b_i \in \mathbb{R}$, and $\boldsymbol{T} = \mathbf{diag}\left(\left[1, \gamma^{\frac{1}{n}}, \gamma^{\frac{2}{n}}, ..., \gamma^{\frac{n-1}{n}}\right]\right)$ is a diagonal matrix that promotes ill-conditioning.

Below, we compare our Algorithm 1 with (i) a fine-tuned fixed stepsize; (ii) a fine-tuned Nesterov's accelerated gradient descent (N-AGD) Nesterov (1983). The tuning is performed on a fine grid with a fixed random number generator, hence shows roughly their best performances. Our parameters are very roughly picked as $\gamma_0 = 1, \tau_1 = \tau_2 = 0.5, T = 1, \bar{f}_0 = 0.1 \cdot f(\boldsymbol{x}^0)$ and no further tuning.

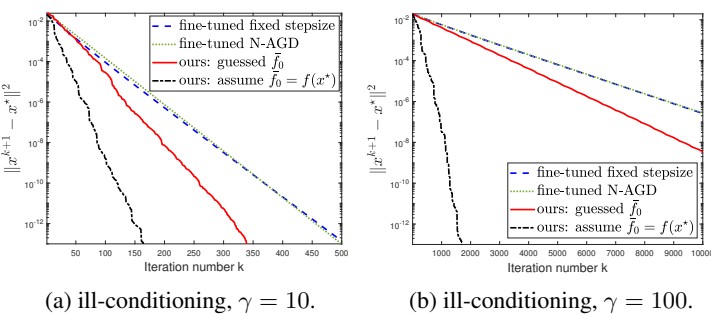

(a) ill-conditioning, $\gamma = 10$.  (b) ill-conditioning, $\gamma = 100$.

Figure 3: Convergence rate comparison, data size $m = 50, n = 10$.

*Remarks* 4.1 (worst-case acceleration). We observe that N-AGD provides almost no acceleration in the ill-conditioning setting here. Let us note that its well-known $O(1/k^2)$ rate is only guaranteed in a worst-case sense and does not necessarily accelerate in practice, see a discussion in (Ryu & Yin, 2022, Sec. 12.3).

*Remarks* 4.2. Due to rough choice of parameters, our guessed $\bar{f}_0 = 0.1 \cdot f(x^0)$ admits a significant performance gap compared to an ideal tune-free case $\bar{f}_0 = f(x^\star)$ (not a priori knowledge). How to improve such a gap is left for future work.

### 4.3 NON-CONVEX MNIST

Here, we consider the MNIST classification problem via a 2-layer neural network, with ReLu activation, 200 hidden units, and softmax loss function. Following the literature, we consider a mini-batch setting. We compare ours with the state-of-the-art algorithms, Nesterov's accelerated gradient descent (N-AGD) Nesterov (1983) and Adaptive moment estimation (Adam) Kingma & Ba (2015).

#### 4.3.1 TUNE-FREE CASE

We start with a special case that stepsize $\alpha_k = f(x^k)/\|\nabla f(x^k)\|^2$ alone works nicely. We consider minimizing the softmax loss only (no regularization) under a relatively large mini-batch size.

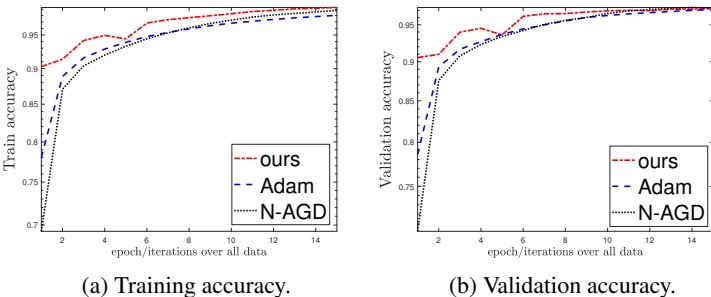

(a) Training accuracy.    (b) Validation accuracy.

Figure 4: Our tune-free case, with mini-batch size 1024.

*Remarks* 4.3. Fig 4a and Fig 4b record the training and validation accuracies, respectively. We observe that they share a highly similar trend (but not the same). Ours exhibits consistent advantages over the others.

*Remarks* 4.4 (parameter details). N-AGD's stepsize is fined-tuned to $1.5 \times 10^{-5}$. Adam has too many hyper-parameters, and is only roughly tuned, with $\alpha = 10^{-3}, \beta_1 = 0.8, \beta_2 = 0.899, \epsilon = 10^{-8}$ (the suggested default has a worse performance in our setting).

#### 4.3.2 GENERAL CASE

Here, we consider a general case, minimizing softmax loss function with $l_2$-norm regularization (on the weights). We adopt a commonly used mini-batch size of 128. Our Algorithm 1 is applied, with roughly picked parameters $\bar{f}_0 = 0, \gamma_0 = 1, T = 5, \tau_1 = 0.25$ ($\tau_2$ omitted).

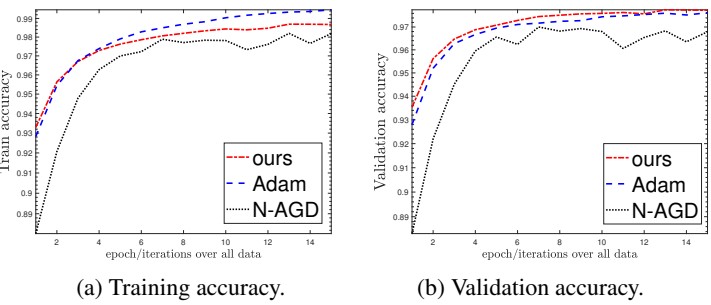

(a) Training accuracy.    (b) Validation accuracy.

Figure 5: General case with $l_2$-norm regularization; mini-batch size 128.

*Remarks* 4.5. Ours only has advantage in the validation stage, where consistently higher accuracy is observed. Luckily, the validation accuracy is all we need, hence ours remains a better choice. Additionally, we suspect our advantage can be enlarged if more careful parameter choices are employed, which is left for future research.

## 5 CONCLUSION

In this work, we established a general theory on the adaptive stepsize selection issue, including feasible selection range, convergence rate, and optimal choice. Specifically, in the convex case, we show an adaptive range $(0, 2\alpha_k^\star)$ that guarantees convergence, which enlarges the classical fixed one $(0, 2/L)$. Its centre $\alpha_k^\star$ is the optimal choice, admitting an exact linear rate with factor $\sin^2 \eta_k$. Our theory also applies to a non-convex function, except the situation is much more challenging. The optimal stepsize can now be negative, and the feasible range set could be empty when some orthogonality arises. On the other hand, if a feasible stepsize choice always exists, then convergence to the global optimal point is guaranteed.

Despite the great power of our theory, it involves some optimal point information. To enable its practical use, we propose an approximation strategy. Such an approximation can be exact in a special practical scenario but in general sub-optimal. It also admits an exact linear convergence rate, and we numerically test its power through several examples. Outstandingly, a tune-free version works nicely for the non-convex MNIST problem via neural networks.

## 6 REPRODUCIBILITY STATEMENT

All figures in this manuscript can be reproduced via the MATLAB codes submitted as supplementary material.

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

## A APPENDIX

The gradient descent (GD) iterates are

$$\boldsymbol{x}^{k+1} = \boldsymbol{x}^k - \alpha_k \nabla f(\boldsymbol{x}^k), \quad k = 0, 1, \dots. \tag{A.1}$$

We assume $\nabla f(\boldsymbol{x}^k) \neq \boldsymbol{0}$, unless $\boldsymbol{x}^k = \boldsymbol{x}^\star$. This assumption is necessary, since otherwise stepsize selection becomes trivial.

### A.1 PROOF OF PROPOSITION 2.1

Our Proposition 2.1, restated here as

**Proposition A.1** (range). *Consider GD in equation A.1. While iterates not converged, let stepsize*

$$\alpha_k \in \left( \frac{2\langle \boldsymbol{x}^\star - \boldsymbol{x}^k, -\nabla f(\boldsymbol{x}^k) \rangle}{\|\nabla f(\boldsymbol{x}^k)\|^2}, 0 \right) \bigcup \left( 0, \frac{2\langle \boldsymbol{x}^\star - \boldsymbol{x}^k, -\nabla f(\boldsymbol{x}^k) \rangle}{\|\nabla f(\boldsymbol{x}^k)\|^2} \right), \quad k = 0, 1, \dots \tag{A.2}$$

*If such $\alpha_k$ exists $\forall k$. Then, convergence to the global optimal point is guaranteed.*

*Proof.* Let us note that

$$
\begin{aligned}
\|\boldsymbol{x}^{k+1} - \boldsymbol{x}^\star\|^2 - \|\boldsymbol{x}^k - \boldsymbol{x}^\star\|^2 &= -\left\|\boldsymbol{x}^{k+1} - \boldsymbol{x}^k\right\|^2 - 2\langle \boldsymbol{x}^\star - \boldsymbol{x}^{k+1}, \boldsymbol{x}^{k+1} - \boldsymbol{x}^k \rangle, \\
&= -(\alpha_k)^2 \|\nabla f(\boldsymbol{x}^k)\|^2 - 2\left\langle \boldsymbol{x}^\star - \boldsymbol{x}^k + \alpha_k \nabla f(\boldsymbol{x}^k), -\alpha_k \nabla f(\boldsymbol{x}^k) \right\rangle, \\
&= \alpha_k^2 \left\|\nabla f(\boldsymbol{x}^k)\right\|^2 - 2\left\langle \boldsymbol{x}^\star - \boldsymbol{x}^k, -\alpha_k \nabla f(\boldsymbol{x}^k) \right\rangle, \\
&= \alpha_k \left( \alpha_k \left\|\nabla f(\boldsymbol{x}^k)\right\|^2 + 2\left\langle \boldsymbol{x}^\star - \boldsymbol{x}^k, -\nabla f(\boldsymbol{x}^k) \right\rangle \right). \tag{A.3}
\end{aligned}
$$

All we need is the above right-hand side being negative, implying the ground-truth error is strictly decreasing, hence guarantees convergence. This yields equation A.2, which involves one empty set, depends on the sign of the term $\langle \boldsymbol{x}^\star - \boldsymbol{x}^k, -\nabla f(\boldsymbol{x}^k) \rangle$. The proof is now concluded. $\square$

### A.2 PROOF OF THEOREM 2.1

Our Theorem 2.1, restated here as

**Theorem A.1** (optimal choice). *Consider GD in equation A.1. The optimal $k$-th choice is given by*

$$\alpha_k^\star = \frac{\langle \boldsymbol{x}^\star - \boldsymbol{x}^k, -\nabla f(\boldsymbol{x}^k) \rangle}{\|\nabla f(\boldsymbol{x}^k)\|^2} = \frac{\|\boldsymbol{x}^\star - \boldsymbol{x}^k\|}{\|\nabla f(\boldsymbol{x}^k)\|} \cos \eta_k, \tag{A.4}$$

*where $\eta_k = \arccos \frac{\langle \boldsymbol{x}^\star - \boldsymbol{x}^k, -\nabla f(\boldsymbol{x}^k) \rangle}{\|\boldsymbol{x}^\star - \boldsymbol{x}^k\| \|\nabla f(\boldsymbol{x}^k)\|}$. It admits the following exact adaptive linear rate:*

$$\|\boldsymbol{x}^{k+1} - \boldsymbol{x}^\star\|^2 = \left(\Pi_{t=0}^k \sin^2 \eta_t\right) \|\boldsymbol{x}^0 - \boldsymbol{x}^\star\|^2, \quad k = 0, 1, \dots. \tag{A.5}$$

*Proof.* Following from equation A.3, we would like its right-hand side term as negative as possible, which leads to

$$\underset{\alpha_k}{\text{minimize}} \ (\alpha_k)^2 \left\|\nabla f(\boldsymbol{x}^k)\right\|^2 - 2\alpha_k \left\langle \boldsymbol{x}^\star - \boldsymbol{x}^k, -\nabla f(\boldsymbol{x}^k) \right\rangle. \tag{A.6}$$

Its solution is

$$\alpha_k^\star = \frac{\langle \boldsymbol{x}^\star - \boldsymbol{x}^k, -\nabla f(\boldsymbol{x}^k) \rangle}{\|\nabla f(\boldsymbol{x}^k)\|^2} = \frac{\|\boldsymbol{x}^\star - \boldsymbol{x}^k\|}{\|\nabla f(\boldsymbol{x}^k)\|} \cos \eta_k, \tag{A.7}$$

where $\eta_k = \arccos \frac{\langle \boldsymbol{x}^\star - \boldsymbol{x}^k, -\nabla f(\boldsymbol{x}^k) \rangle}{\|\boldsymbol{x}^\star - \boldsymbol{x}^k\| \|\nabla f(\boldsymbol{x}^k)\|}$. Substituting it back to equation A.6, we obtain the minimal objective value being

$$(\alpha_k^\star)^2 \left\|\nabla f(\boldsymbol{x}^k)\right\|^2 - 2\alpha_k^\star \left\langle \boldsymbol{x}^\star - \boldsymbol{x}^k, -\nabla f(\boldsymbol{x}^k) \right\rangle = -\|\boldsymbol{x}^\star - \boldsymbol{x}^k\|^2 \cos^2 \eta_k. \tag{A.8}$$

At last, by equation A.3, we obtain

$$
\begin{aligned}
\|\boldsymbol{x}^{k+1} - \boldsymbol{x}^\star\|^2 - \|\boldsymbol{x}^k - \boldsymbol{x}^\star\|^2 &= -\|\boldsymbol{x}^\star - \boldsymbol{x}^k\|^2 \cos^2 \eta_k, \\
\iff \|\boldsymbol{x}^{k+1} - \boldsymbol{x}^\star\|^2 &= \sin^2 \eta_k \|\boldsymbol{x}^\star - \boldsymbol{x}^k\|^2. \tag{A.9}
\end{aligned}
$$

The proof is concluded by considering all iterations, from 0 to the current $k$-th one. $\square$

### A.3 PROOF OF PROPOSITION 2.2

**Lemma A.1** (Baillon-Haddad Theorem). *Let function $f : \mathbb{R}^n \to \mathbb{R}$ be $L$-smooth. The following holds:*

$$\frac{1}{L}\|\nabla f(\boldsymbol{x}) - \nabla f(\boldsymbol{y})\|^2 \leq \langle \boldsymbol{x} - \boldsymbol{y}, \nabla f(\boldsymbol{x}) - \nabla f(\boldsymbol{y}) \rangle, \quad \forall \boldsymbol{x}, \boldsymbol{y} \in \mathbb{R}^n. \tag{A.10}$$

Our Proposition 2.2, restated here as

**Proposition A.2.** *Suppose function $f : \mathbb{R}^n \to \mathbb{R}$ is $L$-smooth. Then,*

$$\alpha_k^\star = \frac{\langle \boldsymbol{x}^\star - \boldsymbol{x}^k, -\nabla f(\boldsymbol{x}^k) \rangle}{\|\nabla f(\boldsymbol{x}^k)\|^2} \geq \frac{1}{L}, \quad k = 0, 1.... \tag{A.11}$$

*Proof.* By Lemma A.1, we have

$$\frac{1}{L}\|\nabla f(\boldsymbol{x}^\star) - \nabla f(\boldsymbol{x}^k)\|^2 \leq \langle \boldsymbol{x}^\star - \boldsymbol{x}^k, \nabla f(\boldsymbol{x}^\star) - \nabla f(\boldsymbol{x}^k) \rangle. \tag{A.12}$$

Rearranging the terms concludes the proof. $\qquad\square$

### A.4 PROOF OF THEOREM 3.1

Our Theorem 3.1, restated here as

**Theorem A.2.** *Consider GD in equation A.1. While iterates not converged, we propose stepsize*

$$\alpha_k^\dagger = \gamma_0 \cdot \frac{f(\boldsymbol{x}^k) - \bar{f}_0}{\|\nabla f(\boldsymbol{x}^k)\|^2}, \tag{A.13}$$

*where $\gamma_0$ is a tunable parameter; $\bar{f}_0$ is a guessed smallest objective value. It admits the following exact linear rate:*

$$\|\boldsymbol{x}^{k+1} - \boldsymbol{x}^\star\|^2 = \left( \Pi_{t=0}^k\ \delta_t \right) \|\boldsymbol{x}^0 - \boldsymbol{x}^\star\|^2, \tag{A.14}$$

*where*

$$\delta_t = 1 - \frac{\gamma_0}{\sigma_t}\left( 2 - \frac{\gamma_0}{\sigma_t} \right)\cos^2\eta_t, \qquad \sigma_t = \frac{\langle \boldsymbol{x}^\star - \boldsymbol{x}^t, -\nabla f(\boldsymbol{x}^t) \rangle}{f(\boldsymbol{x}^t) - \bar{f}_0}, \tag{A.15}$$

*and where $\eta_t = \arccos \frac{\langle \boldsymbol{x}^\star - \boldsymbol{x}^t, -\nabla f(\boldsymbol{x}^t) \rangle}{\|\boldsymbol{x}^\star - \boldsymbol{x}^t\|\|\nabla f(\boldsymbol{x}^t)\|}$.*

*Proof.* Recall error characterization from equation A.3

$$\|\boldsymbol{x}^{k+1} - \boldsymbol{x}^\star\|^2 - \|\boldsymbol{x}^k - \boldsymbol{x}^\star\|^2 = (\alpha_k)^2 \left\| \nabla f(\boldsymbol{x}^k) \right\|^2 - 2\alpha_k \left\langle \boldsymbol{x}^\star - \boldsymbol{x}^k, -\nabla f(\boldsymbol{x}^k) \right\rangle. \tag{A.16}$$

Substituting $\alpha_k^\dagger$ in equation A.13 to the right-hand side above, yields

$$
\begin{aligned}
\text{r.h.s.} &= \left( \gamma_0 \cdot \frac{f(\boldsymbol{x}^k) - \bar{f}_0}{\|\nabla f(\boldsymbol{x}^k)\|^2} \right)^2 \left\| \nabla f(\boldsymbol{x}^k) \right\|^2 - 2\gamma_0 \cdot \frac{f(\boldsymbol{x}^k) - \bar{f}_0}{\|\nabla f(\boldsymbol{x}^k)\|^2} \left\langle \boldsymbol{x}^\star - \boldsymbol{x}^k, -\nabla f(\boldsymbol{x}^k) \right\rangle, \\
&= \left( \gamma_0 \cdot \frac{f(\boldsymbol{x}^k) - \bar{f}_0}{\|\nabla f(\boldsymbol{x}^k)\|} \right)^2 - 2\gamma_0 \cdot \frac{f(\boldsymbol{x}^k) - \bar{f}_0}{\|\nabla f(\boldsymbol{x}^k)\|^2} \left\langle \boldsymbol{x}^\star - \boldsymbol{x}^k, -\nabla f(\boldsymbol{x}^k) \right\rangle, \\
&= \left( \frac{\langle \boldsymbol{x}^\star - \boldsymbol{x}^k, -\nabla f(\boldsymbol{x}^k) \rangle}{\|\nabla f(\boldsymbol{x}^k)\|} \right)^2 \left( \left( \gamma_0 \cdot \frac{f(\boldsymbol{x}^k) - \bar{f}_0}{\langle \boldsymbol{x}^\star - \boldsymbol{x}^k, -\nabla f(\boldsymbol{x}^k) \rangle} \right)^2 - \right. \\
&\qquad \left. 2\gamma_0 \cdot \frac{f(\boldsymbol{x}^k) - \bar{f}_0}{\langle \boldsymbol{x}^\star - \boldsymbol{x}^k, -\nabla f(\boldsymbol{x}^k) \rangle} \right),
\end{aligned} \tag{A.17}
$$

Let $\sigma_k = \frac{\langle \boldsymbol{x}^\star - \boldsymbol{x}^k, -\nabla f(\boldsymbol{x}^k) \rangle}{f(\boldsymbol{x}^k) - \bar{f}_0}$. Invoke the l.h.s. of equation A.16, we arrive at

$$
\begin{aligned}
\|\boldsymbol{x}^{k+1} - \boldsymbol{x}^\star\|^2 - \|\boldsymbol{x}^k - \boldsymbol{x}^\star\|^2 &= \left( \frac{\langle \boldsymbol{x}^\star - \boldsymbol{x}^k, -\nabla f(\boldsymbol{x}^k) \rangle}{\|\nabla f(\boldsymbol{x}^k)\|} \right)^2 \left( \left( \frac{\gamma_0}{\sigma_k} \right)^2 - 2 \cdot \frac{\gamma_0}{\sigma_k} \right), \\
&= \left( \frac{\langle \boldsymbol{x}^\star - \boldsymbol{x}^k, -\nabla f(\boldsymbol{x}^k) \rangle}{\|\nabla f(\boldsymbol{x}^k)\| \|\boldsymbol{x}^k - \boldsymbol{x}^\star\|} \right)^2 \left( \left( \frac{\gamma_0}{\sigma_k} \right)^2 - 2 \cdot \frac{\gamma_0}{\sigma_k} \right) \|\boldsymbol{x}^k - \boldsymbol{x}^\star\|^2, \\
&= \cos^2 \eta_k \cdot \left( \left( \frac{\gamma_0}{\sigma_k} \right)^2 - 2 \cdot \frac{\gamma_0}{\sigma_k} \right) \|\boldsymbol{x}^k - \boldsymbol{x}^\star\|^2, \quad (A.18)
\end{aligned}
$$

where $\eta_k = \arccos \frac{\langle \boldsymbol{x}^\star - \boldsymbol{x}^k, -\nabla f(\boldsymbol{x}^k) \rangle}{\|\boldsymbol{x}^\star - \boldsymbol{x}^k\| \|\nabla f(\boldsymbol{x}^k)\|}$. It follows that

$$
\|\boldsymbol{x}^{k+1} - \boldsymbol{x}^\star\|^2 = \left( 1 - \frac{\gamma_0}{\sigma_k} \left( 2 - \frac{\gamma_0}{\sigma_k} \right) \cos^2 \eta_k \right) \|\boldsymbol{x}^k - \boldsymbol{x}^\star\|^2. \quad (A.19)
$$

Considering all iterations $t = 0, 1, 2...k$ gives equation A.14. The proof is now concluded. $\square$

### A.5 PROOF OF PROPOSITION 3.1

Our Proposition 3.1, restated here as

**Proposition A.3** (tune-free stepsize). *Consider GD in equation 2.2. Suppose function $f$ is convex, with optimal objective value $f(\boldsymbol{x}^\star)$ known in advance. Then, any choice from below*

$$
\widetilde{\alpha}_k = \gamma_0 \cdot \frac{f(\boldsymbol{x}^k) - f(\boldsymbol{x}^\star)}{\|\nabla f(\boldsymbol{x}^k)\|^2}, \quad \gamma_0 \in (0, 2]. \quad (A.20)
$$

*guarantees convergence, with an exact linear rate:*

$$
\|\boldsymbol{x}^{k+1} - \boldsymbol{x}^\star\|^2 = \left( \Pi_{t=0}^k \; \delta_t \right) \|\boldsymbol{x}^0 - \boldsymbol{x}^\star\|^2, \quad (A.21)
$$

*where*

$$
\delta_t = 1 - \frac{\gamma_0}{\sigma_t} \left( 2 - \frac{\gamma_0}{\sigma_t} \right) \cos^2 \eta_t, \quad \sigma_t = \frac{\langle \boldsymbol{x}^\star - \boldsymbol{x}^t, -\nabla f(\boldsymbol{x}^t) \rangle}{f(\boldsymbol{x}^t) - f(\boldsymbol{x}^\star)}, \quad (A.22)
$$

*where $\eta_t = \arccos \frac{\langle \boldsymbol{x}^\star - \boldsymbol{x}^t, -\nabla f(\boldsymbol{x}^t) \rangle}{\|\boldsymbol{x}^\star - \boldsymbol{x}^t\| \|\nabla f(\boldsymbol{x}^t)\|}$.*

*Proof.* Given $\boldsymbol{x}^k \neq \boldsymbol{x}^\star$, the following holds:

$$
f(\boldsymbol{x}^\star) > f(\boldsymbol{x}^k) + \langle \boldsymbol{x}^\star - \boldsymbol{x}^k, \nabla f(\boldsymbol{x}^k) \rangle, \quad (A.23)
$$

where we exclude the case of a linear function, since it is unbounded below and is therefore trivial to minimize. Rearranging the terms, yields

$$
f(\boldsymbol{x}^k) - f(\boldsymbol{x}^\star) < \langle \boldsymbol{x}^\star - \boldsymbol{x}^k, -\nabla f(\boldsymbol{x}^k) \rangle \quad (A.24)
$$

Suppose $\gamma_0 \in (0, 2]$. Then,

$$
\gamma_0 \cdot \frac{f(\boldsymbol{x}^k) - f(\boldsymbol{x}^\star)}{\|\nabla f(\boldsymbol{x}^k)\|^2} < \frac{2 \langle \boldsymbol{x}^\star - \boldsymbol{x}^k, -\nabla f(\boldsymbol{x}^k) \rangle}{\|\nabla f(\boldsymbol{x}^k)\|^2}. \quad (A.25)
$$

It says that the left-hand side above always lies within the feasible range, recall equation A.2. Its convergence rate follows directly from Theorem A.2. The proof is now concluded. $\square$

### A.6 PROOF OF PROPOSITION 3.2

Our Proposition 3.2, restated here as

**Proposition A.4.** *Suppose function $f : \mathbb{R}^n \to \mathbb{R}$ is $L$-smooth. Then,*

$$
\frac{f(\boldsymbol{x}^k) - f(\boldsymbol{x}^\star)}{\|\nabla f(\boldsymbol{x}^k)\|^2} \geq \frac{1}{2L}, \quad (A.26)
$$

*Proof.* The $L$-smoothness assumption implies

$$f(\boldsymbol{y}) \leq f(\boldsymbol{x}) + \langle \boldsymbol{y} - \boldsymbol{x}, \nabla f(\boldsymbol{x}) \rangle + \frac{L}{2} \|\boldsymbol{y} - \boldsymbol{x}\|^2, \quad \forall \boldsymbol{x}, \boldsymbol{y} \in \mathbb{R}^n. \tag{A.27}$$

We may perform the following minimization:

$$\underset{\boldsymbol{y}}{\text{minimize}} \; f(\boldsymbol{x}) + \langle \boldsymbol{y} - \boldsymbol{x}, \nabla f(\boldsymbol{x}) \rangle + \frac{L}{2} \|\boldsymbol{y} - \boldsymbol{x}\|^2, \tag{A.28}$$

and obtain a minimizer

$$\widehat{\boldsymbol{y}} = \boldsymbol{x} - \frac{1}{L} \nabla f(\boldsymbol{x}) \tag{A.29}$$

Substituting it back, yields

$$f(\widehat{\boldsymbol{y}}) \leq f(\boldsymbol{x}) - \frac{1}{2L} \|\nabla f(\boldsymbol{x})\|^2, \quad \forall \boldsymbol{x} \in \mathbb{R}^n. \tag{A.30}$$

It follows that

$$f(\boldsymbol{x}^\star) \leq f(\widehat{\boldsymbol{y}}) \leq f(\boldsymbol{x}^k) - \frac{1}{2L} \|\nabla f(\boldsymbol{x}^k)\|^2. \tag{A.31}$$

Rearranging the terms concludes the proof. $\qquad\square$

A.7  PROOF OF PROPOSITION 3.3

Our Proposition 3.3, restated here as

**Proposition A.5** (optimality gap). *Let function $f : \mathbb{R}^n \to \mathbb{R}$ be $L$-smooth. Then,*

$$\underbrace{\frac{\langle \boldsymbol{x}^\star - \boldsymbol{x}^k, -\nabla f(\boldsymbol{x}^k) \rangle}{\|\nabla f(\boldsymbol{x}^k)\|^2}}_{optimal} - \underbrace{\frac{f(\boldsymbol{x}^k) - f(\boldsymbol{x}^\star)}{\|\nabla f(\boldsymbol{x}^k)\|^2}}_{estimated\,(\gamma_0 = 1)} \geq \frac{1}{2L}. \tag{A.32}$$

*Proof.* The proof follows instantly from (Nesterov, 2018, Theorem 2.1.5)

$$f(\boldsymbol{x}^\star) \geq f(\boldsymbol{x}^k) + \langle \boldsymbol{x}^\star - \boldsymbol{x}^k, \nabla f(\boldsymbol{x}^k) \rangle + \frac{1}{2L} \|\nabla f(\boldsymbol{x}^\star) - \nabla f(\boldsymbol{x}^k)\|^2. \tag{A.33}$$

Dividing both sides with $\|\nabla f(\boldsymbol{x}^k)\|^2$ (non-zero by assumption) concludes the proof. $\qquad\square$