# R^2 quadratic example, Figure 1

This code reproduces our Figure 1.

```matlab
clear;clc;
rng('default')
    max_itr = 1e2;
      gamma = 10;      %condition number
          L = gamma; %Lipschitz constant
      x_ini = [L;1]; %initialization
% 1. Exact line search
      x_line = x_ini;
for   k = 1 : max_itr
       x(1) = L*((L-1)/(L+1))^k;
       x(2) = (-(L-1)/(L+1))^k;
      %record
      x_line = [x_line,x'];
err_line(k) = norm(x - 0)^2;
end
% 2. Ours
       x_our = x_ini;
for   k = 1 : max_itr
         k1 = floor((k+1)/2);
         k2 = floor(k/2);

    factor  = ((L-1)/2)^k1 * ((L-1)/(L^2+1))^k2;
        x(1) =  L^(k2+1-k1) * factor;
        x(2) = (-1)^(k1+k2) * factor;
       %record
       x_our = [x_our,x'];
 err_our(k) = norm(x - 0)^2;
end
% convergence rate
figure
semilogy(err_line,'-','LineWidth',3); hold on
semilogy(err_our,'--r','LineWidth',3);
xlabel('Iteration number k','FontSize',15)
ylabel('$$ \Vert x^k - x^\star \Vert^2 $$',...
        'Interpreter','latex','FontSize',20)
legend('exact line search','ours', 'FontSize',22)
axis([1 max_itr 1e-25 inf])
% trajectory
figure
f = @(x1,y1) 1/2*(x1.^2 + L*y1.^2);
fcontour(f,[-13 13  -5 5],'--k')
hold on
p1 = plot(x_line(1,:),x_line(2,:),'-o','LineWidth',2);
p2 = plot(x_our(1,:),x_our(2,:),'r-.o','LineWidth',2);
legend([p1 p2],'exact line search','ours','FontSize',22)
xlabel('$$  x_1  $$','Interpreter','latex','FontSize',15)
ylabel('$$  x_2  $$','Interpreter','latex','FontSize',15)
```

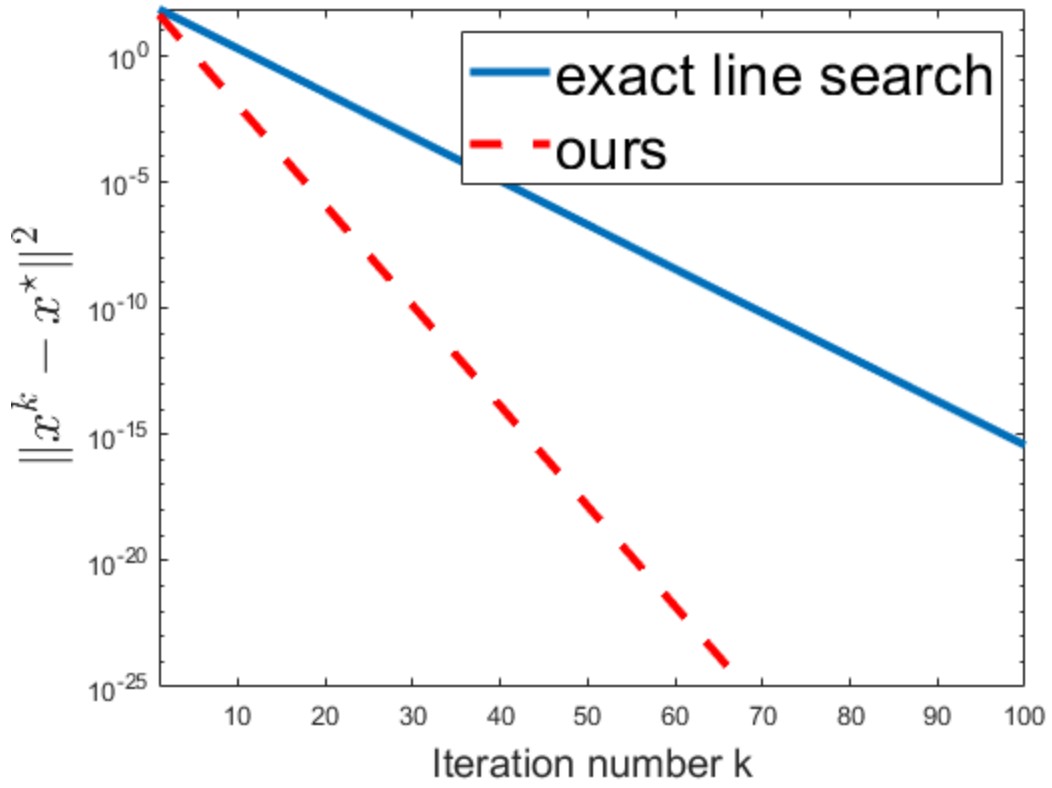

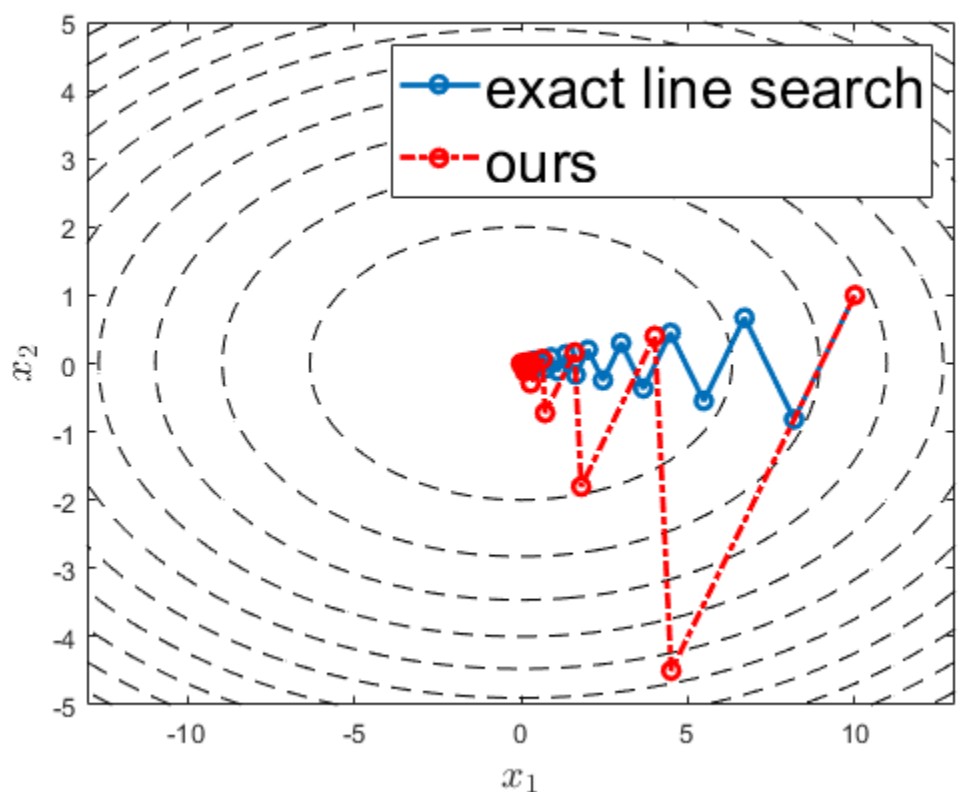

*Published with MATLAB® R2020b*

# R^2 quadratic example, instant convergence

This code reproduces our Figure 2.

```matlab
clear;clc;
rng('default')
max_itr = 1e2;
  gamma = 10;      %condition number
      L = gamma; %Lipschitz constant
  x_ini = [50;0]; %initialization
      A = diag([1,sqrt(L)]);
% 1. ours
      x = x_ini;
 x_ours = x;
for   k = 1 : max_itr
     dx = A'*A*x ;
  alpha = norm(A*x)^2 / norm(dx)^2;
      x = x - alpha*dx;
      %record
 x_ours = [x_ours,x];
end
% 2. fixed stepsize 1/L
      x = x_ini;
    x_L = x;
for   k = 1 : max_itr
     dx = A'*A*x ;
      x = x - 1/L*dx;
      %record
    x_L = [x_L,x];
end
% trajectory 1
figure
f = @(x1,y1) 1/2*(x1.^2 + L*y1.^2);
fcontour(f,[0 51  -5 5],'--k')
hold on
p1 = plot(x_ours(1,:),x_ours(2,:),'r-o','LineWidth',1.2);
legend(p1 ,'ours, 1 step','FontSize',32)
xlabel('$$  x_1  $$','Interpreter','latex','FontSize',15)
ylabel('$$  x_2  $$','Interpreter','latex','FontSize',15)
% trajectory 2
figure
f = @(x1,y1) 1/2*(x1.^2 + L*y1.^2);
fcontour(f,[0 51  -5 5],'--k')
hold on
p1 = plot(x_L(1,:),x_L(2,:),'b-o','LineWidth',1.2);
legend(p1 ,'stepsize 1/L','FontSize',32)
xlabel('$$  x_1  $$','Interpreter','latex','FontSize',15)
ylabel('$$  x_2  $$','Interpreter','latex','FontSize',15)
```

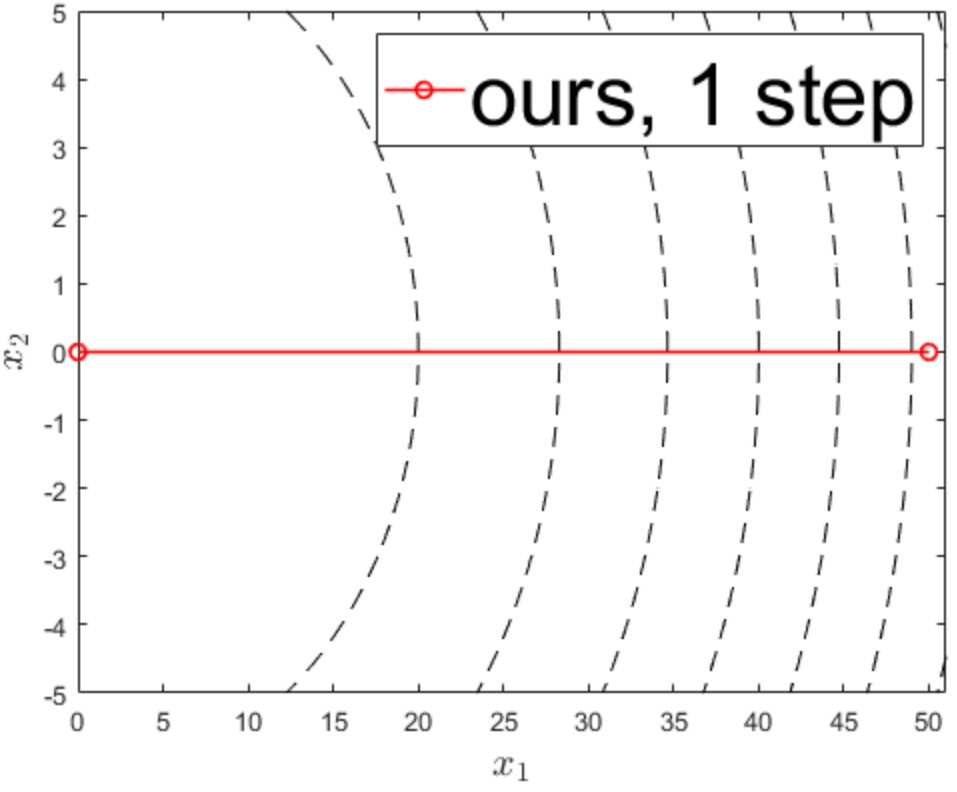

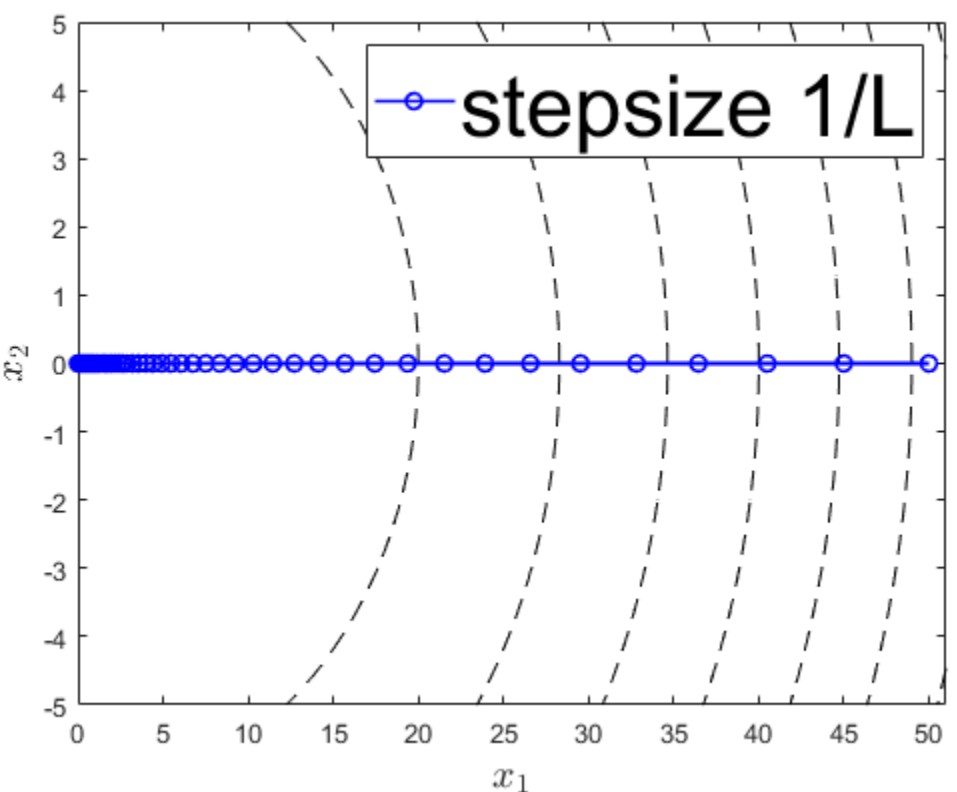

# Geometric Program, Figure 3a

This code reproduces our Figure 3a.

```matlab
clear;clc;
rng('default')
max_itr = 5e2;
      m = 50;
      n = 10;
      A = randn(m,n);
      b = randn(m,1);
  x_ini = zeros(n,1); %initialization
% ill-conditioning via a diognal matrix T
   gamma = 10;
     T(1) = 1;
for    i = 1:n-1
  T(i+1) = gamma^(i/n);
end
      A = A.*T;
% 1. constant stepsize, fine-tuned
      x = x_ini;
for    k = 1 : max_itr
      dx = A'*(1./(ones(m,1)'*exp(A*x + b)).*exp(A*x + b) ) ;
      x = x - 3.4e-2 * dx;

  x_L{k} =  x;
end
% 2. Nesterov's AGD, fine-tuned
      x = x_ini;
   alpha = 0;
    beta = 0;
   x_old = x;
for    k = 1 : max_itr
      x = x + beta * (x - x_old);
      dx = A'*(1./(ones(m,1)'*exp( A*x + b)).*exp( A*x + b) ) ;
   x_old = x;
      x = x - 1.72e-2 * dx;
   % momentum parameters
   a_old = alpha;
   alpha = (1 + sqrt(1+4*a_old^2))/2;
   beta  = (a_old - 1) / alpha;

x_AGD{k} = x;
end
% 3. Ours, Algorithm 1
      x = x_ini;
     f0 = 0.1*log(ones(m,1)'*exp(A*x + b)); %guessed
   tau1 = 0.5; % not sensitive, e.g., 0.6, 0.7
   tau2 = 0.5;
 gamma0 = 1;
for    k = 1 : max_itr
      dx = A'*(1./(ones(m,1)'*exp( A*x + b)).*exp(A*x + b)) ;
```

```
        f(k) = log(ones(m,1)'*exp(A*x + b));
         a_k = gamma0*(f(k) - f0)/norm(dx)^2 ;

      if a_k > 0
          x2 = x - a_k * dx;
      f(k+1) = log(ones(m,1)'*exp( A*x2 + b));
          if f(k+1) > f(k)
             gamma0 = tau1 * gamma0;
                 x2 = x;
              f(k+1) = f(k);
          end
      else
          f0 = tau2 * f0;
       f(k+1) = f(k);
      end
          x = x2;

x_our1{k} = x;
end
```

# ground-truth error via CVX

```
% CVX package
cvx_begin quiet
cvx_precision best
    variables x_cvx(n,1)
    minimize ( log( ones(m,1)'*exp(A*x_cvx + b) )  )
cvx_end
% 4. Ours, assume accurate f0
    obj_opt = log(ones(m,1)'*exp(A*x_cvx  + b));
         x = x_ini;
for       k = 1 : max_itr
        dx = A'*(1./(ones(m,1)'*exp( A*x + b)).*exp( A*x + b) ) ;
         f = log( ones(m,1)'*exp( A*x + b) );
      a_k = (f - obj_opt)/norm(dx)^2 ;
         x = x - a_k * dx;

  x_our2{k} = x;
end
% ground-truth error
for   i = 1 : max_itr
   err_L(i) = norm(x_L{i} - x_cvx)^2/n;
 err_AGD(i) = norm(x_AGD{i} - x_cvx)^2/n;
err_our1(i) = norm(x_our1{i} - x_cvx)^2/n;
err_our2(i) = norm(x_our2{i} - x_cvx)^2/n;
end
figure
semilogy(err_L,'--b','LineWidth',2); hold on
semilogy(err_AGD,':','LineWidth',2,'Color',[0.47,0.67,0.19]);
semilogy(err_our1,'-r','LineWidth',2);
semilogy(err_our2,'-.k','LineWidth',2);
xlabel('Iteration number k','FontSize',15)
ylabel('$$ \Vert x^{k+1} - x^\star \Vert^2 $$',...
```

```
                'Interpreter','latex','FontSize',20)
   legend('fine-tuned fixed stepsize', 'fine-tuned N-AGD',...
    'ours: guessed $$ \bar{f}_0 $$',...
    'ours: assume $$\bar{f}_0 = f(x^\star)$$',...
    'Interpreter','latex','FontSize',18)
   axis([1 inf 1e-13 inf])
```

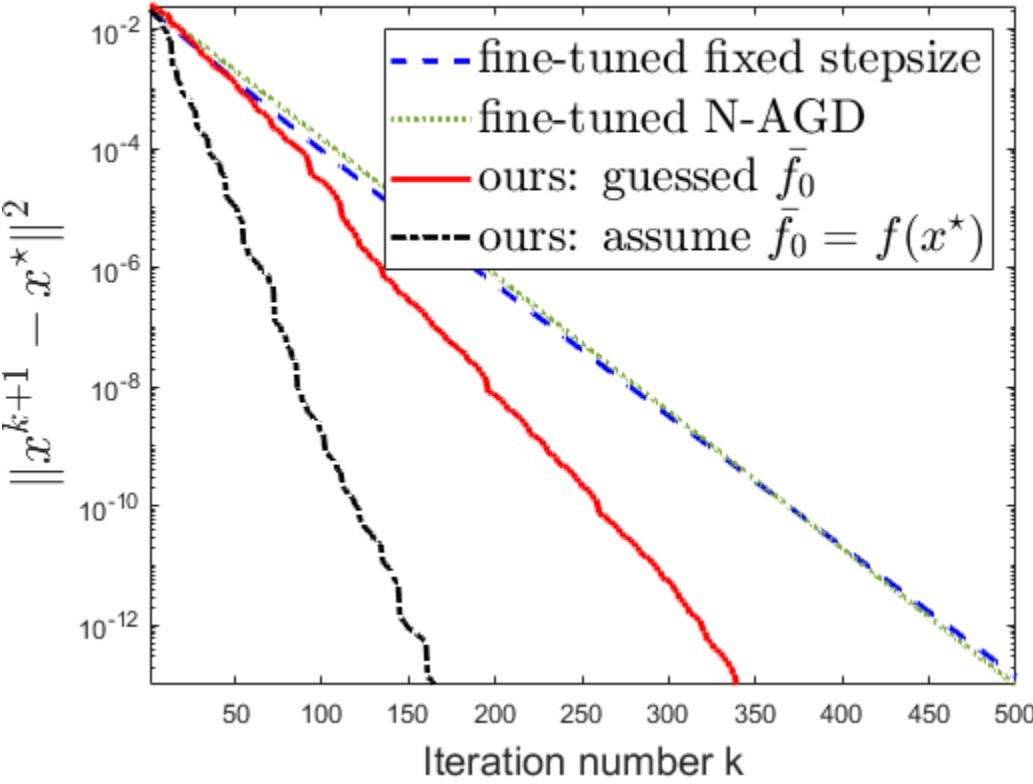

*Published with MATLAB® R2020b*

# Geometric Program, Figure 3b

This code reproduces our Figure 3b.

```matlab
clear;clc;
rng('default')
max_itr = 1e4;
      m = 50;
      n = 10;
      A = randn(m,n);
      b = randn(m,1);
  x_ini = zeros(n,1); %initialization
% ill-conditioning via a diognal matrix T
   gamma = 100;
     T(1) = 1;
for    i = 1:n-1
  T(i+1) = gamma^(i/n);
end
      A = A.*T;
% 1. constant stepsize, fine-tuned
      x = x_ini;
for    k = 1 : max_itr
      dx = A'*(1./(ones(m,1)'*exp(A*x + b)).*exp(A*x + b) ) ;
      x = x - 6.5e-4 * dx;

  x_L{k} =  x;
end
% 2. Nesterov's AGD, fine-tuned
      x = x_ini;
   alpha = 0;
    beta = 0;
   x_old = x;
for    k = 1 : max_itr
      x = x + beta * (x - x_old);
      dx = A'*(1./(ones(m,1)'*exp( A*x + b)).*exp( A*x + b) ) ;
   x_old = x;
      x = x - 3.25e-4 * dx;
   % momentum parameters
   a_old = alpha;
   alpha = (1 + sqrt(1+4*a_old^2))/2;
   beta  = (a_old - 1) / alpha;

x_AGD{k} = x;
end
% 3. Ours, Algorithm 1
      x = x_ini;
      f0 = 0.1*log(ones(m,1)'*exp(A*x + b)); %guessed
    tau1 = 0.5; % not sensitive, e.g., 0.6, 0.7
    tau2 = 0.5;
  gamma0 = 1;
for    k = 1 : max_itr
      dx = A'*(1./(ones(m,1)'*exp( A*x + b)).*exp(A*x + b)) ;
```

```matlab
        f(k) = log(ones(m,1)'*exp(A*x + b));
         a_k = gamma0*(f(k) - f0)/norm(dx)^2 ;

    if a_k > 0
        x2 = x - a_k * dx;
    f(k+1) = log(ones(m,1)'*exp( A*x2 + b));
        if f(k+1) > f(k)
           gamma0 = tau1 * gamma0;
               x2 = x;
           f(k+1) = f(k);
        end
    else
         f0 = tau2 * f0;
     f(k+1) = f(k);
    end
         x = x2;

x_our1{k} = x;
end
```

# ground-truth error via CVX

```matlab
% CVX package
cvx_begin quiet
cvx_precision best
    variables x_cvx(n,1)
    minimize ( log( ones(m,1)'*exp(A*x_cvx + b) )  )
cvx_end

% 4. Ours, assume accurate f0
    obj_opt = log(ones(m,1)'*exp(A*x_cvx  + b));
         x = x_ini;
for       k = 1 : max_itr
         dx = A'*(1./(ones(m,1)'*exp( A*x + b)).*exp( A*x + b) ) ;
          f = log( ones(m,1)'*exp( A*x + b) );
        a_k = (f - obj_opt)/norm(dx)^2 ;
          x = x - a_k * dx;

  x_our2{k} = x;
end
% ground-truth error
for   i = 1 : max_itr
   err_L(i) = norm(x_L{i} - x_cvx)^2/n;
 err_AGD(i) = norm(x_AGD{i} - x_cvx)^2/n;
err_our1(i) = norm(x_our1{i} - x_cvx)^2/n;
err_our2(i) = norm(x_our2{i} - x_cvx)^2/n;
end

figure
semilogy(err_L,'--b','LineWidth',2); hold on
semilogy(err_AGD,':','LineWidth',2,'Color',[0.47,0.67,0.19]);
semilogy(err_our1,'-r','LineWidth',2);
semilogy(err_our2,'-.k','LineWidth',2);
```

```
xlabel('Iteration number k','FontSize',15)
ylabel('$$ \Vert x^{k+1} - x^\star \Vert^2 $$',...
        'Interpreter','latex','FontSize',20)
legend('fine-tuned fixed stepsize', 'fine-tuned N-AGD',...
 'ours: guessed $$ \bar{f}_0 $$',...
 'ours: assume $$\bar{f}_0 = f(x^\star)$$',...
 'Interpreter','latex','FontSize',18)
 axis([1 inf 1e-13 inf])
```

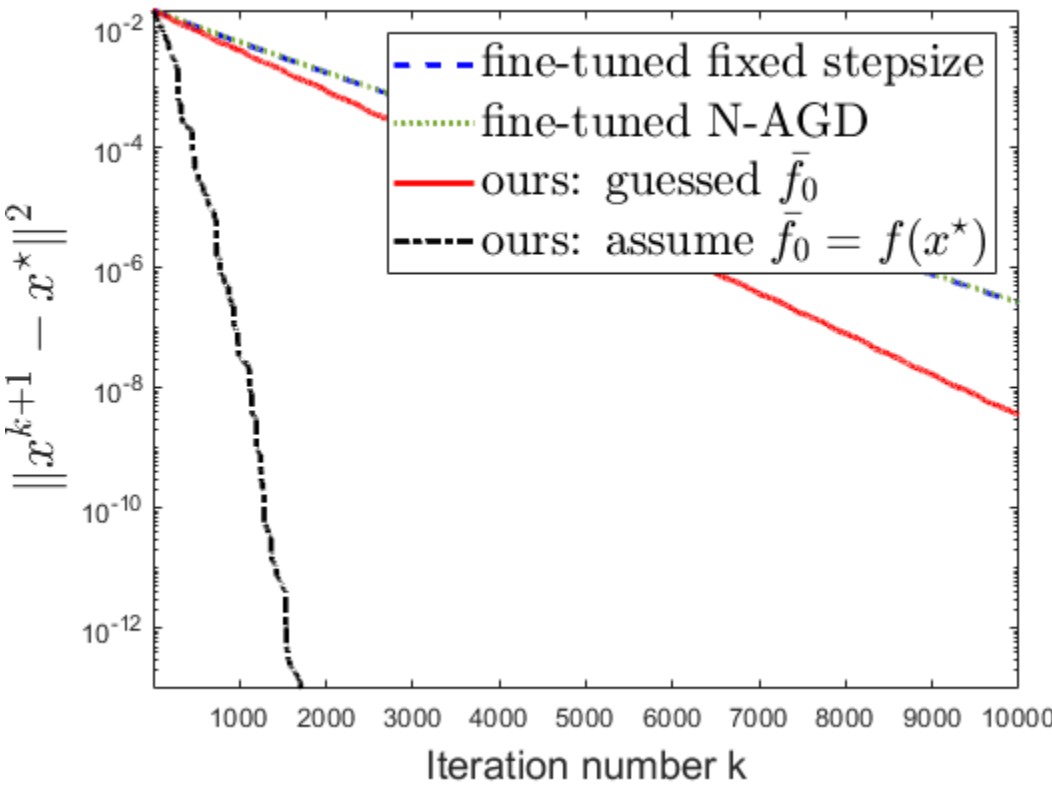

# Non-convex MINIST, tune-free case

This code reproduces our Figure 4. Our stepsize is tune-free here.

```matlab
clear;clc;
% train data
 t_data = load('mnist_train.csv');
  t_lab = t_data(:,1);
      Y = zeros(10,length(t_data));
for   i = 1:length(t_data)
      Y(t_lab(i)+1,i) = 1;
end
% validation data
 v_data = load('mnist_test.csv');
  v_lab = v_data(:,1);
% normalization
   t_im = t_data(:,2:785);
   t_im = t_im/255;
   v_im = v_data(:,2:785);
   v_im = v_im/255;
% data matrices
X_train = [t_im,ones(length(t_data),1)];
 X_test = [v_im,ones(length(v_data),1)];

% GD optimizer
rng('default')
    b_s = 1024;                 % mini-batch size
    eps = 15;                   % epoch number
      D = 28^2;
      K = 10;
      H = 200;
 W1_ini = 0.01*randn(D+1,H+1); % initialization
 W2_ini = 0.01*randn(H+1,K);   % initialization
 itr_in = floor(length(t_data)/b_s) ;

% 1. Ours, tune-free:
        W1 = W1_ini;
        W2 = W2_ini;
for    eps = 1:eps
    for  i = 1:itr_in
         X = X_train(1+(i-1)*b_s:i*b_s, :);
        Lb = Y(:, 1+(i-1)*b_s : i*b_s)';
       h_l = max(0, X * W1);
         s = h_l * W2;
     exp_s = exp(s);
     probs = exp_s ./ sum(exp_s, 2) ;
        ds = probs - Lb;
       dW2 = h_l' * ds  ;
        dh = ds * W2';
   dh(h_l <= 0) = 0;
       dW1 = X' * dh ;
         f = sum(- Lb.* log(probs),'all');
```

```matlab
        % GD
        W2 = W2 - f/norm(dW2,'fro')^2*dW2;
        W1 = W1 - f/norm(dW1,'fro')^2*dW1;
    end
    % train accuracy
        h_l = max(0, X_train * W1);
         s = h_l * W2;
     exp_s = exp(s);
     probs = exp_s ./ sum(exp_s, 2) ;
    [~,I0] = max(probs,[],2);
    Train_ours(eps) = mean(I0 == t_lab+1);
    % validation accuracy
        h_l = max(0, X_test * W1);
     s_test = h_l * W2 ;
     exp_st = exp(s_test);
     prob_t = exp_st ./ sum(exp_st, 2) ;
    [~,I_t] = max(prob_t,[],2);
    Val_ours(eps) = mean(I_t == v_lab+1);
end

% 2. Adam
        W1 = W1_ini;
        W2 = W2_ini;
    dW1_a1 = zeros(D+1,H+1);
    dW1_a2 = zeros(D+1,H+1);
    dW2_a1 = zeros(H+1,K);
    dW2_a2 = zeros(H+1,K);
     beta1 = 0.8;
     beta2 = 0.899;
     alpha = 1e-3;
for    eps = 1:eps
    for  i = 1:itr_in
         X = X_train(1+(i-1)*b_s:i*b_s, :);
        Lb = Y(:, 1+(i-1)*b_s : i*b_s )';
      h_l = max(0, X * W1);
         s = h_l * W2;
     exp_s = exp(s);
     probs = exp_s ./ sum(exp_s, 2) ;
        ds = probs - Lb;
       dW2 = h_l' * ds  ;
        dh = ds * W2';
    dh(h_l <= 0) = 0;
       dW1 = X' * dh ;

       itr = i+(eps-1)*itr_in;
    dW1_a1 = beta1 * dW1_a1 + (1-beta1) * dW1;
    dW1_a1 = dW1_a1 / (1 -  beta1.^itr);
    dW1_a2 = beta2 * dW1_a2 + (1-beta2) * dW1.^2;
    dW1_a2 = dW1_a2 / (1 -  beta2.^itr);

    dW2_a1 = beta1 * dW2_a1 + (1-beta1) * dW2;
    dW2_a1 = dW2_a1 / (1 -  beta1^itr);
    dW2_a2 = beta2 * dW2_a2 + (1-beta2) * dW2.^2;
    dW2_a2 = dW2_a2 / (1 -  beta2^itr);
```

```matlab
        % GD
        W1 = W1 - alpha ./ (sqrt(dW1_a2) + 1e-8) .* dW1_a1;
        W2 = W2 - alpha ./ (sqrt(dW2_a2) + 1e-8) .* dW2_a1;
    end
    % train accuracy
        h_l = max(0, X_train * W1);
          s = h_l * W2;
     exp_s = exp(s);
     probs = exp_s ./ sum(exp_s, 2) ;
    [~,I0] = max(probs,[],2);
    Train_Adam(eps) = mean(I0 == t_lab+1);
    % validation
        h_l = max(0, X_test * W1);
    s_test = h_l * W2 ;
    exp_st = exp(s_test);
    prob_t = exp_st ./ sum(exp_st, 2) ;
    [M,I_t] = max(prob_t,[],2);
    Val_Adam(eps) = mean(I_t == v_lab+1);
end

% 3. N-AGD
        W1 = W1_ini;
        W2 = W2_ini;
    W2_old = W2;
    W1_old = W1;
     alpha = 0;
      beta = 0;
for    eps = 1:eps
    for  i = 1:itr_in
         X = X_train(1+(i-1)*b_s:i*b_s, :);
        Lb = Y(:, 1+(i-1)*b_s : i*b_s )';
    W2_tmp = W2 + beta * (W2 - W2_old);
    W1_tmp = W1 + beta * (W1 - W1_old);
        h_l = max(0, X * W1_tmp);
          s = h_l * W2_tmp;
     exp_s = exp(s);
     probs = exp_s ./ sum(exp_s, 2) ;
        ds = probs - Lb;
       dW2 = h_l' * ds  ;
        dh = ds * W2_tmp';
  dh(h_l <= 0) = 0;
       dW1 = X' * dh ;
    W2_old = W2;
    W1_old = W1;
     % momentum parameters
     a_old = alpha;
     alpha = (1 + sqrt(1+4*a_old^2))/2;
     beta  = (a_old - 1) / alpha;
        % GD
        W2 = W2_tmp - 1.5e-5*dW2;
        W1 = W1_tmp - 1.5e-5*dW1;
    end
    % train accuracy
        h_l = max(0, X_train * W1);
```

```matlab
        s = h_l * W2;
     exp_s = exp(s);
     probs = exp_s ./ sum(exp_s, 2) ;
    [~,I0] = max(probs,[],2);
    Train_AGD(eps) = mean(I0 == t_lab+1);
    % validation accuracy
       h_l = max(0, X_test * W1);
    s_test = h_l * W2 ;
    exp_st = exp(s_test);
    prob_t = exp_st ./ sum(exp_st, 2) ;
    [~,I_t] = max(prob_t,[],2);
   Val_AGD(eps) = mean(I_t == v_lab+1);
end
% Figures
% (a) Train accuracy plot
figure
semilogy(Train_ours,'-.r','LineWidth',2);hold on;
semilogy(Train_Adam,'--b','LineWidth',2);
semilogy(Train_AGD,  ':k','LineWidth',2);

xlabel('epoch/iterations over all data',...
       'Interpreter','latex','FontSize',15)
ylabel('Train accuracy','Interpreter','latex','FontSize',20)
legend('ours','Adam','N-AGD','FontSize',26)
  axis tight
% (b) Validation accuracy plot
figure
semilogy(Val_ours,'-.r','LineWidth',2);hold on;
semilogy(Val_Adam,'--b','LineWidth',2);
semilogy(Val_AGD,':k','LineWidth',2);

xlabel('epoch/iterations over all data',...
       'Interpreter','latex','FontSize',15)
ylabel('Validation accuracy','Interpreter','latex','FontSize',20)
legend('ours','Adam','N-AGD','FontSize',26)
  axis  tight
```

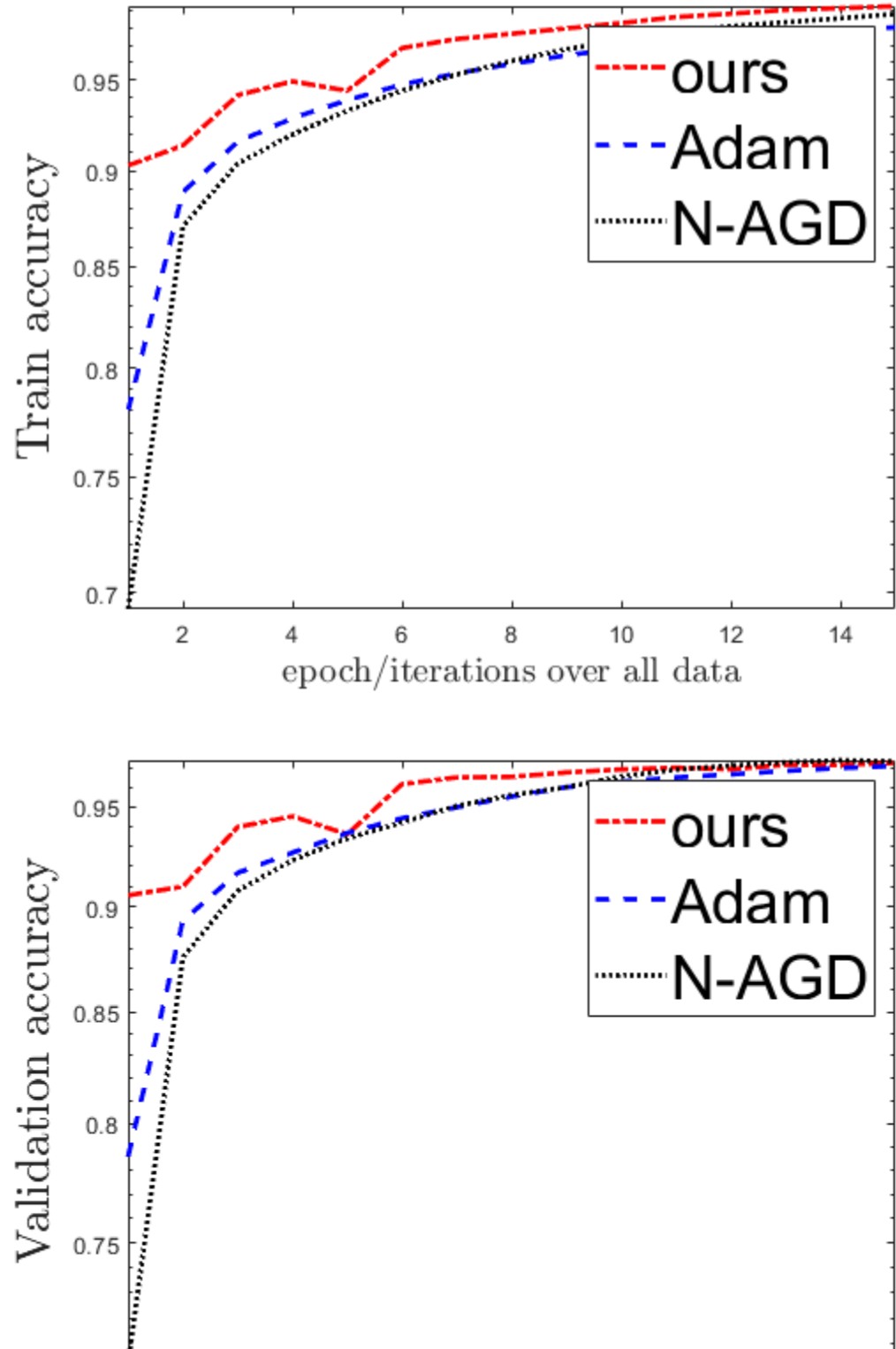

*Published with MATLAB® R2020b*

# Non-convex MINIST, general case

This code reproduces our Figure 5. Our advantage is on the validation accuracy.

```matlab
clear;clc;
% train data
 t_data = load('mnist_train.csv');
  t_lab = t_data(:,1);
      Y = zeros(10,length(t_data));
for   i = 1:length(t_data)
      Y(t_lab(i)+1,i) = 1;
end
% validation data
 v_data = load('mnist_test.csv');
  v_lab = v_data(:,1);
% normalization
   t_im = t_data(:,2:785);
   t_im = t_im/255;
   v_im = v_data(:,2:785);
   v_im = v_im/255;
% data matrices
X_train = [t_im,ones(length(t_data),1)];
 X_test = [v_im,ones(length(v_data),1)];

% GD optimizer
rng('default')
    b_s = 128;                 % mini-batch size
    eps = 15;                  % epoch number
     mu = 1e-3;                % regularization
      D = 28^2;
      K = 10;
      H = 200;
 itr_in = floor(length(t_data)/b_s);
 W1_ini = 0.01*randn(D+1,H+1); % initialization
 W2_ini = 0.01*randn(H+1,K);   % initialization

% 1. Ours, Algorithm1:
        W1 = W1_ini;
        W2 = W2_ini;
    gamma0 = 1;
      tau1 = 0.25;
         T = 5;
for    eps = 1:eps
    for  i = 1:itr_in
         X = X_train(1+(i-1)*b_s:i*b_s, :);
        Lb = Y(:, 1+(i-1)*b_s : i*b_s )';
       h_l = max(0, X * W1);
         s = h_l * W2;
     exp_s = exp(s);
     probs = exp_s ./ sum(exp_s, 2) ;
        ds = probs - Lb;
       dW2 = h_l' * ds + mu*W2;
```

```matlab
        dh = ds * W2';
    dh(h_l <= 0) = 0;
        dW1 = X' * dh + mu*W1;
      f(i) = sum(- Lb.* log(probs),'all')...
            + mu/2*(norm(W1,'fro')^2 + norm(W2,'fro')^2);
     % GD
    W2_new = W2 - gamma0*f(i)/norm(dW2,'fro')^2*dW2;
    W1_new = W1 - gamma0*f(i)/norm(dW1,'fro')^2*dW1;

     % correction
       h_l = max(0, X * W1_new);
         s = h_l * W2_new;
     exp_s = exp(s);
     probs = exp_s ./ sum(exp_s, 2) ;
    f(i+1) = sum(- Lb.* log(probs),'all')...
            + mu*1/2*(norm(W1,'fro')^2 + norm(W2,'fro')^2);
     if f(i+1) > T * f(i)
     gamma0 = tau1 * gamma0;
     W2_new = W2;
     W1_new = W1;
     f(i+1) = f(i);
     end

        W2 = W2_new;
        W1 = W1_new;
     end
     % train accuracy
       h_l = max(0, X_train * W1);
         s = h_l * W2;
     exp_s = exp(s);
     probs = exp_s ./ sum(exp_s, 2) ;
    [~,I0] = max(probs,[],2);
    Train_ours(eps) = mean(I0 == t_lab+1);
     % validation accuracy
       h_l = max(0, X_test * W1);
     s_test = h_l * W2;
     exp_st = exp(s_test);
     prob_t = exp_st ./ sum(exp_st, 2) ;
    [~,I_t] = max(prob_t,[],2);
    Val_ours(eps) = mean(I_t == v_lab+1);
end

% 2. Adam
        W1 = W1_ini;
        W2 = W2_ini;
    dW1_a1 = zeros(D+1,H+1);
    dW1_a2 = zeros(D+1,H+1);
    dW2_a1 = zeros(H+1,K);
    dW2_a2 = zeros(H+1,K);
     beta1 = 0.8;
     beta2 = 0.899;
     alpha = 1e-3;
for     eps = 1:eps
    for  i = 1:itr_in
```

```matlab
       X = X_train(1+(i-1)*b_s:i*b_s, :);
      Lb = Y(:, 1+(i-1)*b_s : i*b_s )';
     h_l = max(0, X * W1);
       s = h_l * W2;
   exp_s = exp(s);
   probs = exp_s ./ sum(exp_s, 2) ;
      ds = probs - Lb;
     dW2 = h_l' * ds + mu*W2;
      dh = ds * W2';
  dh(h_l <= 0) = 0;
     dW1 = X' * dh + mu*W1;

     itr = i+(eps-1)*itr_in;
  dW1_a1 = beta1 * dW1_a1 + (1-beta1) * dW1;
  dW1_a1 = dW1_a1 / (1 -  beta1.^itr);
  dW1_a2 = beta2 * dW1_a2 + (1-beta2) * dW1.^2;
  dW1_a2 = dW1_a2 / (1 -  beta2.^itr);

  dW2_a1 = beta1 * dW2_a1 + (1-beta1) * dW2;
  dW2_a1 = dW2_a1 / (1 -  beta1^itr);
  dW2_a2 = beta2 * dW2_a2 + (1-beta2) * dW2.^2;
  dW2_a2 = dW2_a2 / (1 -  beta2^itr);
      % GD
      W1 = W1 - alpha ./ (sqrt(dW1_a2) + 1e-8) .* dW1_a1;
      W2 = W2 - alpha ./ (sqrt(dW2_a2) + 1e-8) .* dW2_a1;
    end
    % train accuracy
     h_l = max(0, X_train * W1);
       s = h_l * W2;
   exp_s = exp(s);
   probs = exp_s ./ sum(exp_s, 2) ;
  [~,I0] = max(probs,[],2);
  Train_Adam(eps) = mean(I0 == t_lab+1);
    % validation
     h_l = max(0, X_test * W1);
  s_test = h_l * W2 ;
  exp_st = exp(s_test);
  prob_t = exp_st ./ sum(exp_st, 2) ;
 [M,I_t] = max(prob_t,[],2);
  Val_Adam(eps) = mean(I_t == v_lab+1);
end

% 3. N-AGD
      W1 = W1_ini;
      W2 = W2_ini;
  W2_old = W2;
  W1_old = W1;
   alpha = 0;
    beta = 0;
for    eps = 1:eps
    for  i = 1:itr_in
        X = X_train(1+(i-1)*b_s:i*b_s, :);
       Lb = Y(:, 1+(i-1)*b_s : i*b_s )';
   W2_tmp = W2 + beta * (W2 - W2_old);
```

```matlab
    W1_tmp = W1 + beta * (W1 - W1_old);
      h_l = max(0, X * W1_tmp);
        s = h_l * W2_tmp;
    exp_s = exp(s);
    probs = exp_s ./ sum(exp_s, 2) ;
       ds = probs - Lb;
      dW2 = h_l' * ds + mu*W2;
      dh = ds * W2_tmp';
  dh(h_l <= 0) = 0;
     dW1 = X' * dh + mu*W1;
  W2_old = W2;
  W1_old = W1;
    % momentum parameters
    a_old = alpha;
    alpha = (1 + sqrt(1+4*a_old^2))/2;
    beta  = (a_old - 1) / alpha;
       % GD
       W2 = W2_tmp - 1e-5*dW2;
       W1 = W1_tmp - 1e-5*dW1;
    end
    % train accuracy
       h_l = max(0, X_train * W1);
         s = h_l * W2;
     exp_s = exp(s);
     probs = exp_s ./ sum(exp_s, 2) ;
    [~,I0] = max(probs,[],2);
    Train_AGD(eps) = mean(I0 == t_lab+1);
    % validation accuracy
       h_l = max(0, X_test * W1);
    s_test = h_l * W2 ;
    exp_st = exp(s_test);
    prob_t = exp_st ./ sum(exp_st, 2) ;
    [~,I_t] = max(prob_t,[],2);
    Val_AGD(eps) = mean(I_t == v_lab+1);
end
% Figures
% (a) Train accuracy plot
figure
semilogy(Train_ours,'-.r','LineWidth',2);hold on;
semilogy(Train_Adam,'--b','LineWidth',2);
semilogy(Train_AGD,  ':k','LineWidth',2);

xlabel('epoch/iterations over all data',...
       'Interpreter','latex','FontSize',15)
ylabel('Train accuracy','Interpreter','latex','FontSize',20)
legend('ours','Adam','N-AGD','FontSize',26)
  axis tight
% (b) Validation accuracy plot
figure
semilogy(Val_ours,'-.r','LineWidth',2);hold on;
semilogy(Val_Adam,'--b','LineWidth',2);
semilogy(Val_AGD,':k','LineWidth',2);

xlabel('epoch/iterations over all data',...
```

```
                    'Interpreter','latex','FontSize',15)
ylabel('Validation accuracy','Interpreter','latex','FontSize',20)
legend('ours','Adam','N-AGD','FontSize',26)
   axis  tight
```

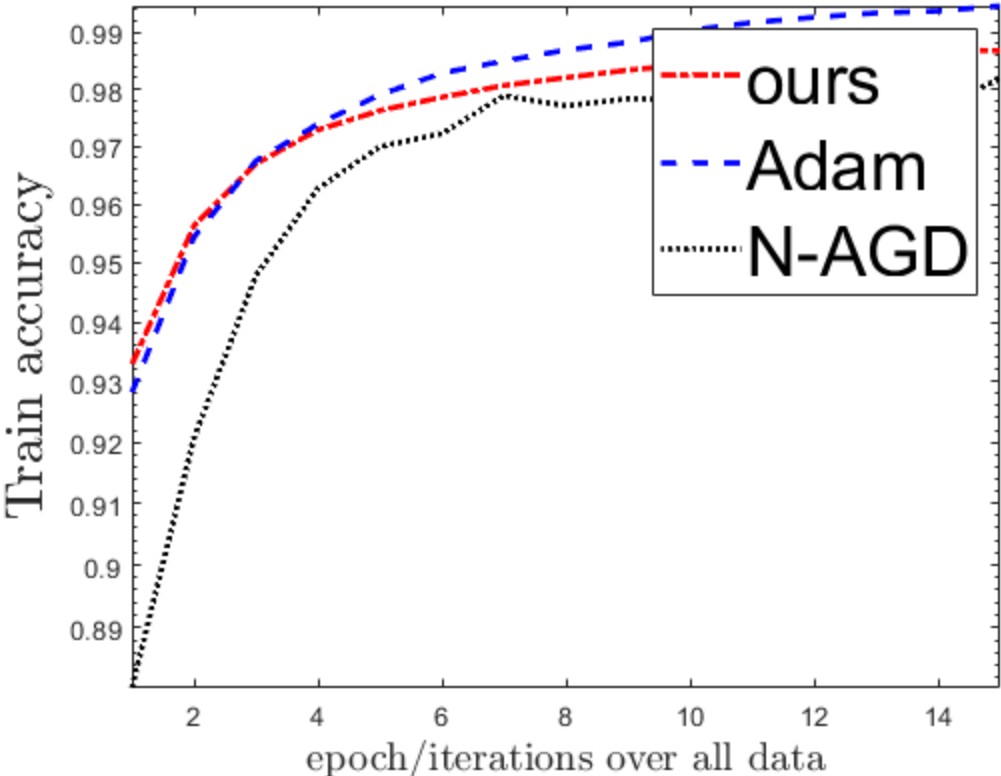

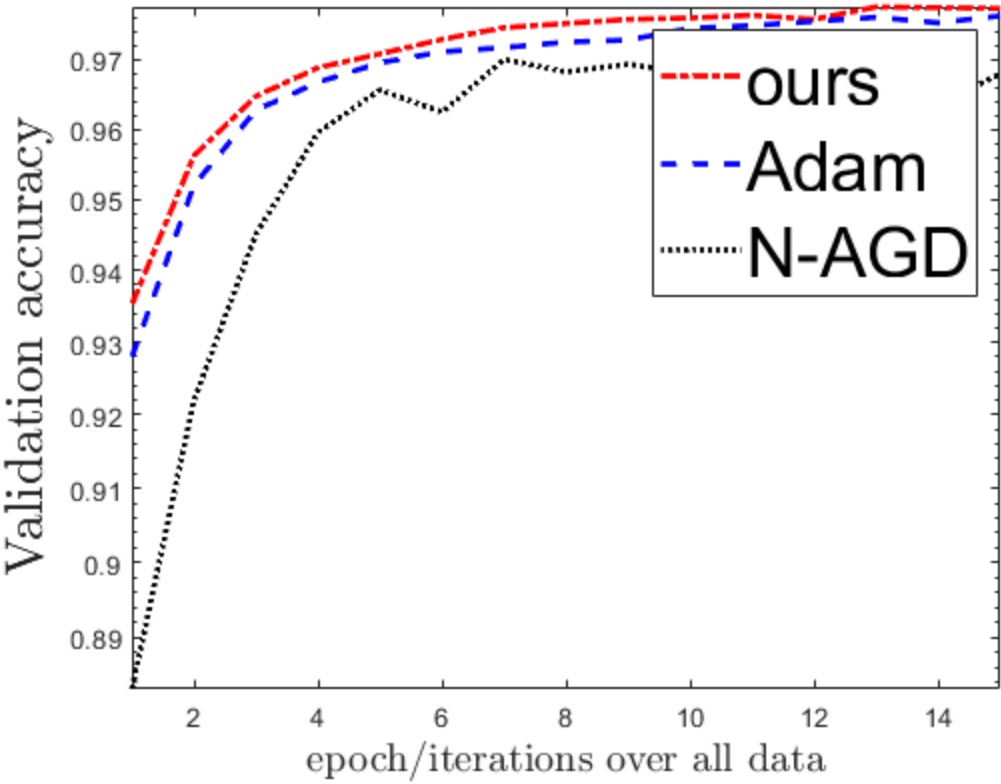

*Published with MATLAB® R2020b*