# OpenReview forum: "Exact linear-rate gradient descent: optimal adaptive stepsize theory and practical use"
_ICLR.cc/2025/Conference — Submitted to ICLR 2025_

### Official Review · Reviewer_1n9o · 2024-10-24

**Soundness:** 1
**Presentation:** 1
**Contribution:** 1
**Rating:** 1
**Confidence:** 5

**Summary:**

The paper proposes an adaptive stepsize selection scheme for gradient descent (GD). The main theoretical contribution is providing an expression for what is claimed to be an optimal stepsize choice, which depends on the (implicitly assumed to be unique) solution to the problem. For practical implementation, they propose approximating this with a Polyak-like stepsize estimating inf_x f(x). The authors provide convergence analysis and some numerical experiments on MNIST and quadratic optimization.

**Strengths:**

- Numerical experiments are performed and their plots are reported

**Weaknesses:**

- The definition of the theoretical stepsize proposed depends on x* but it's not clear that x* is unique or that this stepsize is well-defined if x* is not unique. No further assumptions on the objective function f are ever stated to ensure uniqueness of x*. No discussion of what will happen if x* is not unique is given.

- The practical use stepsize given is just a Polyak stepsize approximating inf f by \bar{f}_0. Yet, no reference to Polyak is made nor to any papers studying the Polyak stepsize and related variants, which are quite numerous. In this way the discussion of related work is severely lacking.

- The quality of writing is far below a level that is acceptable for publication. Many statements are mathematically incomplete (e.g., line 155 and many others) or outright incorrect (e.g., the Baillon-Hadad theorem on line 650). Many statements have implicit assumptions that are never stated and not always satisfied or verifiable (e.g., line 146 and many others). None of the convergence results make sense mathematically as there is no reason for x* to be unique - how can \|x_k-x*\|^2 go to 0 for two different x*?

- There is no comparison of the tuning-free algorithm to other tuning-free gradient descent algorithms, of which there is a significant body of work.

**Questions:**

- Why are there no citations to relevant works on Polyak stepsize and tuning-free methods?
- What are the assumptions made on f for each of the results, and do they depend on x* being unique?
- Can you actually verify the assumptions you make on alpha_k in any way if you know in advance f or at least properties that it satisfies, e.g., Lipschitz-smoothness or gradient domination?

---

> ### Comment · Reviewer_1n9o · 2024-11-26
>
> The authors have failed to respond regarding the mathematical errors of their work that I pointed out (e.g., uniqueness of the solution). For this reason I maintain my current score.

---

### Official Review · Reviewer_szwm · 2024-11-03

**Soundness:** 3
**Presentation:** 2
**Contribution:** 2
**Rating:** 5
**Confidence:** 3

**Summary:**

This paper proposes a new adaptive stepsize for gradient descent that achieves exact linear convergence rates for convex optimization.

The key contribution is a novel stepsize formula based on the gradient and objective function.

The authors provide two versions of the stepsize: a theoretical version and a practical version.

They demonstrate the efficacy of this approach through some preliminary examples.

**Strengths:**

**S1:** The paper is well-written and easy to follow, with a clear presentation of the introduction and background on line-search-free first-order methods.

**S2:** This paper proves a simple line-search-free variant of gradient descent to minimize smooth convex functions. The proposed stepsize can be dynamically adjusted to capture the curvature information of the problem, allowing for faster convergence.

**S3:** The paper provides a rigorous proof of the linear convergence rate under the convex settings.

**S4:** The paper includes empirical comparisons with other popular optimizers, such as Adam and N-AGD.

**Weaknesses:**

**W1.** The theoretical analysis relies on strong assumptions, namely that the objective function is convex and the optimal objective value $f(x^*)$ is known.

**W2.** The only practical solution proposed in this paper is Algorithm 1. However, the authors do not provide a theoretical analysis for it. In particular, does Algorithm 1 converge in convex settings? What is its iteration complexity in an ergodic sense when the objective function is convex and non-convex?

**W3.** Additional detailed discussion and analysis are necessary and would be beneficial to further clarify and present Algorithm 1.
1. For example, the auto-correction mechanism in Algorithm 1 explicitly requires $g(x) \geq 0$; otherwise, $\overline{f}_0$ may not serve as a reliable estimate of $f(x^*)$.
2. Taking the least squares problem in Problem (3.16) as an example, when $\alpha >0$ and $\alpha \approx0$, Algorithm 1 could get stuck at a point that is neither a local nor a global minimum, as the second correction in Line 322 is never invoked. This can result in a less accurate estimation of $f(x^*)$.

**W4.** Other issues:

1) The proposed algorithm is only suitable for deterministic optimization problems, as it requires calculating the objective function value, making it incompatible with stochastic optimization models. Comparing it with stochastic optimizers like ADAM may be unfair, as ADAM is designed for stochastic settings while the proposed method is deterministic.

2) It would be beneficial for the authors to include comparisons with other leading deterministic algorithms, such as AdaGrad-Norm (AdaGrad stepsizes: Sharp convergence over nonconvex landscapes, JMLR 2020), APGM (Adaptive Proximal Gradient Methods Are Universal Without Approximation, ICML 2024), and AdaBB (Adaptive Barzilai-Borwein method for convex optimization, 2024).

**Questions:**

**Q1.** Could the authors provide theoretical analysis (e.g., oracle or iteration complexity) for the proposed adaptive stepsize strategy in the case where $f(x)$ is non-convex?

**Q2.** The authors mention using a commonly adopted mini-batch size of 128. Is this setting specific to ADAM? The proposed method may not directly extend to stochastic settings if it requires a dynamic estimation of $f(x^*)$.

---

> ### Comment · Reviewer_szwm · 2024-11-26
>
> I have implemented the practical version of the proposed algorithm and found that it converges very fast.
>
> I encourage the authors to carry out a in-depth analysis on this algorithm.
>
> However, I maintain my original score considering a number of concerns put forward by the reviewers.

---

### Official Review · Reviewer_Les3 · 2024-11-04

**Soundness:** 2
**Presentation:** 2
**Contribution:** 2
**Rating:** 1
**Confidence:** 4

**Summary:**

The paper considers selecting a stepsize for gradient descent, in particular when we cannot compute global quantities like smoothness parameters. Though there has been considerable work, including recently, on adaptive step size selection methods such as Adagrad, this paper takes a different view. The idea is to approximate the a step size that looks a lot like the Polyak step size, by quantities that can be estimated (the Polyak step size requires knowing f(x*)).

They use this step size on various experiments, including on the non-convex problem of training a 2 layer MLP.

**Strengths:**

The paper considers a significant and important problem.
The problem is of current interest -- there are papers appearing about related topics every year.
The proposal of a new step size is related to well studied step sizes like Polyak step size, but it seems to have some novel aspects.

**Weaknesses:**

The writing could be significantly improved. There are many examples where the writing deviates from grammatical English, even from the very beginning of the paper. For instance, “due to quantity is not a priori knowledge.” — lines 75-76. In some places this does not impede the understandability of the paper, but in others the problems with the writing indeed make it hard to properly understand what the paper is about, and what its contributions are.

The introduction is generally loose and imprecise, in areas where it should be specifying exactly what the area of contribution is, precisely because this is such a well-researched area. For example, the paper says that though there are several adaptive algorithms implemented and available, “an adaptive stepsize theory has not been established.” This is confusing, since there are many theoretical papers about AdaGrad and other adaptive step size schedules in the last few years in ML and Optimization venues (not to mention that it is also a fairly classical topic).

Then we are told that their optimal stepwise yields a linear rate with factor sin^2 \eta_k — but we do not know what \eta_k is at this point in the paper. They they gone on to say that the theory applies to non-convex functions, but we are not told what is guaranteed in this case. At least an informal statement should be made explaining what is happening, if the authors wish to talk about it directly.

Proposition 2.1 says it guarantees convergence to a global optimum of GD, yet does not require in the statement that the function being optimized be convex. The proof also does not mention convexity, and indeed does not prove anything about global convergence.

In line 146, the paper says that they assume that the gradient is non-zero unless GD has already converged; but then they say that this means that it has converged to x*, but which I understand that the assumption is that they assume they are minimizing a function that has no stationary points other than the unique global optimum.

The experiments are also not particularly convincing. They need to better point to where the weaknesses are of other related methods, where this approach succeeds.

**Questions:**

What are the weakest assumptions that are required about the function f, in order for you to guarantee your results hold?

What is the relationship to the Polyak step size (e.g., paper by Hazan and Kakade)?

---

> ### Comment · Reviewer_Les3 · 2024-12-02
> **Response to rebuttal**
>
> I appreciate the authors' rebuttal, including the fact that they were previously unaware of the literature on Polyak Stepsize.
>
> Overall, there are still numerous parts of this paper that are not explained, and questions posed by myself and the other reviewers that have not been answered in the response.

---

### Official Review · Reviewer_LAqD · 2024-11-05

**Soundness:** 2
**Presentation:** 2
**Contribution:** 2
**Rating:** 3
**Confidence:** 4

**Summary:**

This paper studies some adaptive size rules for smooth functions, including some theoretical optimal ones and practical approximations. Experiments show some advantages of these rules.

**Strengths:**

This paper uses several examples to demonstrate the benefits of using the proposed step size rules.

**Weaknesses:**

1. This paper lacks a comprehensive comparisons with prior art. A similar rule to the proposed step size rule is studied at least in reference [1] and revisited in [2]. For example, in [2], algorithm 2 is similar to the algorithm for practical use in this paper, and to compare rates with [2], can the authors provide details on how to interpret the term $\Pi_{t = 0}^k \delta_t$ in equation 3.2?
2. The optimal choice is not shown to be optimal in detail and not fully understandable to me, i.e., in what sense this choice if optimal, does it achieve fastest global convergence rate or fastest one-step descent?
3. The experiments in Figure 3 rely on a good guess of $\bar{f}_0$, and this introduces another parameter for a step size rule designed for tuning free case.

[1]. Boris T. Polyak. Introduction to optimization. Optimization Software, Inc., New York, 1987.
[2]. Hazan, Elad, and Sham Kakade. "Revisiting the Polyak step size." arXiv preprint arXiv:1905.00313 (2019).

Based on these weakness, I think this paper can be significantly enhanced by a thorough comparison with related works and detailed explanations of the improved convergence rates.

**Questions:**

1. Does Theorem 2.1 and Corollary 2.2 assumes $L$-smoothness?

---

### Author Response · Authors · 2024-11-16
**Main response: prior art clarification**

We thank the reviewers for spending valuable time reviewing our paper.
# Clarification on prior art
We are completely not aware of Polyak's stepsize. **We rediscovered it**. This is why reviewers find that ---  *`looks a lot like Polyak's stepsize, but seems to have some novel aspects'.*

**We guarantee all results in our paper are independently developed from scratch.** After reviewers pointed  out Polyak's stepsize, we found 7 related papers [1-7], and ours turns out only has a single overlap with them in a special case. Other results and our motivation all appear to be significantly different (or completely new), detailed below:

- Overlap:
  - Polyak's stepsize coincides with our tune-free case in eq.(1.6).
- Differences:
  - (i)  Our theoretical choice, $\alpha_k^\star  =\frac{ \Vert  \bf{x}^\star - \bf{x}^{k}   \Vert }{ \Vert   \nabla f (	{\bf{x}}^{k} )   \Vert} \cos\eta_k $ in **eq. (2.5)**, is not found anywhere else.
  - (ii)  All our convergence rates are exact, including $\Pi_{t = 0}^k sin^2 \eta_k $ in **eq. (2.6)** and  $\Pi_{t = 0}^k \delta_t$ in **eq. (3.2)**, which we cannot find anywhere else.
  - (iii)  An adaptive selection range $ (0, 2 \alpha_k^\star) $  in **eq. (2.8)** for convex problems, which is a natural extension of the classical $ (0, 2/L) $, is not found anywhere else.
  - (iv) Our result applies to a non-convex function, with an extended selection range $(2 \alpha_k^\star, 0)$  or $(0, 2 \alpha_k^\star)  $ in **eq. (2.3)**,  is not found anywhere else.
  - (v)  Polyak's stepsize requires $f(x^\star)$, or at least a good lower bound. We proposed an auto-correction procedure in **Algorithm 1** to alleviate this bottleneck, applicable to both convex and non-convex problems, as in Sec. 4.2 and Sec. 4.3.2, with codes open-sourced in the supplementary material. These are not found anywhere else.
  - (vi) We proved that our optimal choice $ \alpha_k^\star$ does not suffer from ill-conditioning  in **Sec. 4.1** (for a well-known 2-d example), and proved a
strictly better performance than the  exact line search  in **Sec. 4.1.1**. These are extremely surprising results, and we have not found anything similar.


- Our motivation:
  - We are motivated by an obstacle --- our theoretical choice $\alpha_k^\star$ is not known in advance. We need a pre-known stepsize choice to satisfy  $\alpha_k \in (0, 2 \alpha_k^\star) $. This motivates a lower bound choice in eq. (1.3).
All papers in [1-7] do not have $ \alpha_k^\star$ or  the feasible range $(0, 2 \alpha_k^\star) $. Things are different in the first place.


### Reference 1

[1] Boris Polyak. Gradient methods for the minimisation of functionals. Computational Mathematics
and Mathematical Physics, 1963.

[2] Boris Polyak. Introduction to optimization. 1987

[3] Nikhil Devanathan and Stephen Boyd. Polyak minorant method for convex optimization. Journal of
Optimization Theory and Applications, 2024.

[4] Xiaoyu Wang, Mikael Johansson, and Tong Zhang. Generalized polyak step size for first order
optimization with momentum, 2023.

[5] Xiaowen Jiang and Sebastian U. Stich. Adaptive SGD with Polyak stepsize and line-search: robust
convergence and variance reduction.  NeurIPS, 2023.

[6] Elad Hazan and Sham Kakade. Revisiting the polyak step size, 2022.

[7] Nicolas Loizou, Sharan Vaswani, Issam Hadj Laradji, and Simon Lacoste-Julien. Stochastic polyak step-size
for SGD: an adaptive learning rate for fast convergence, AISTATS 2021.

## Our efforts for the  literature review
For related work,  apart from the stepsize papers already cited in our Sec. 1.1,
 we have checked the following fabulous optimization textbooks:  Boyd Stephen and Lieven [1], Nesterov [2], Ryu and Yin [3]. In addition, we have  checked  dozens of university lecture notes on gradient descent.
Unfortunately, **none of them mentioned 'Polyak's stepsize'** (many other works from Prof. Polyak are mentioned, but not the stepsize).

**Admittedly, it is quite hard to find this prior art, unless knowing the exact name 'Polyak's stepsize'.**

Prof. Polyak is one of the greatest founders of the optimization field, it would be our honor to have a chance to cite his work.
But again, we are completely not aware of Polyak's stepsize, which was published in 1963.
We sense that the power of 'Polyak's stepsize' might be underestimated. If lucky, our paper might bring some extra revived interest.


### Reference 2
[1] Boyd Stephen and Vandenberghe Lieven. Convex Optimization. Cambridge University Press,  2004.

[2] Yurii Nesterov. Lectures on Convex Optimization. Springer, 2018.

[3] Ernest K Ryu and Wotao Yin. Large-scale convex optimization: algorithms & analyses via monotone operators. Cambridge University Press, 2022.

---

> ### Comment · Reviewer_LAqD · 2024-11-22
>
> Thanks the authors for the responses. However, there are no explanations to my questions, and I still think this paper can be significantly improved in terms of showing explicit convergence rate improvement.

---

### Meta-Review · Area_Chair_oVXi · 2024-12-13

**Metareview:**

Despite the achievement of rediscovering the Polyak stepsize without knowing it before, it is quite clear that this paper does not offer any new insights of this celebrated method. All reviewers agree - and I join them - that this paper should be rejected.

**Additional Comments On Reviewer Discussion:**

see metareview

---

### Decision · Program_Chairs · 2025-01-22

Reject